

# Evaluation of GPM-DPR precipitation estimates with WegenerNet gauge data

Martin Lasser[1,*], Sungmin O[1,2,†], and Ulrich Foelsche[1,2,3]

[1]Institute for Geophysics, Astrophysics, and Meteorology/Institute of Physics (IGAM/IP), NAWI Graz, University of Graz, Austria
[2]FWF-DK Climate Change, University of Graz, Austria
[3]Wegener Center for Climate and Global Change (WEGC), University of Graz, Austria
[*]Now at Astronomical Institute, University of Bern, Bern, Switzerland
[†]Now at Biogeochemical Integration, Max Planck Institute for Biogeochemistry, Jena, Germany

**Correspondence:** Ulrich Foelsche (ulrich.foelsche@uni-graz.at)

**Abstract.** The core satellite of the Global Precipitation Measurement (GPM) mission provides precipitation observations measured with the Dual frequency Precipitation Radar (DPR). The precipitation can only be estimated from the radar data, and therefore, independent validations using direct precipitation observation on the ground as a true reference need to be performed. Moreover, the quality and the accuracy of the measurements depend on various influencing factors. In this way, a validation

may help to minimise those uncertainties. The DPR provides three different radar rain rate estimates for the GPM core satellite: Ku-band-only rain rates, Ka-band-only rain rates and a product combining the two frequencies. This study presents an evaluation of the three GPM-DPR surface precipitation estimates based on the gridded precipitation data of the WegenerNet, a local scale terrestrial network of 153 meteorological stations in southeast Austria.

The validation is based on a graphical and a statistical approach using only data where both Ku- and Ka-band measurements are

available. The data delivered from the WegenerNet are gauge-based gridded rainfall observations; the meteorological winter is excluded due to technical reasons. The focus lies on the resemblance of the variability within the whole network and the over- and underestimation of the precipitation through the GPM-DPR. During the last four years 22 rainfall events were observed by the GPM-DPR over the WegenerNet and the analysis rests upon these rainfall events. The WegenerNet provides a large number of gauges within each GPM-DPR footprint. Its biases are well studied and corrected, thus, it can be taken as a robust ground

reference. This work also includes considerations on the limits of such comparisons between small terrestrial networks with a high density of stations and precipitation observations from a satellite.

Our results show that the GPM-DPR estimates basically match with the WegenerNet measurements, but absolute quantities are biased. The three types of radar estimates deliver similar results, where Ku-band and dual frequency estimates are very close to each other. On a general level, Ka-band precipitation estimates deliver the best results due to the high number of light rainfall

events.



## 1 Introduction

The Global Precipitation Measurement (GPM) mission aims to give consistent and comprehensive information about Earth's global precipitation. The mission is led by the National Aeronautics and Space Administration (NASA) and the Japan Aerospace and Exploration Agency (JAXA). It is the successor mission of the Tropical Rainfall Measurement Mission (TRMM) and tar-

gets to provide advanced information on rain and snow characteristics from multi-satellites. It measures fundamental quantities of the global water cycle, such as the precipitation amount, on a global level. The results are utilised in weather forecasts, flood predictions and river managements, studies on global change and climate variations as well as the assessment of the global water cycle (see JAXA, 2017)

The GPM Core Observatory (GPM-CO) satellite is equipped with an active Dual frequency Precipitation Radar (DPR) and

a passive microwave imager (GPM microwave imager - GMI). Together with a constellation of partner satellites from international space and weather agencies, such as the National Oceanic and Atmospheric Administration (NOAA), the Centre national d'études spatiales (CNES) and the Indian Space Research Organisation (ISRO) or the European Organisation for the Exploitation of Meteorological Satellites (EUMETSAT), it provides global-scale precipitation data. The GPM-CO satellite was launched in 2014 and flies at an altitude of 407 kilometers in a non-Sun-synchronous orbit that covers the Earth from 65°S

to 65°N. Next to its own measurements, it serves as a reference for unifying the data from the partner satellites (Skofronick-Jackson et al., 2016). The main instrument is in principle a weather surveillance radar operating on two frequencies to map weather events across its swath. The two frequencies allow to estimate the sizes of precipitation particles and detect a wider range of precipitation rates. The microwave imagers augment the core satellite and enable a high temporal resolution for global precipitation maps.

In this study solely the surface rain rate estimates derived from DPR on board the GPM-CO are evaluated and compared to the rain-gauge-based gridded data from the WegenerNet. The WegenerNet is a local scale dense terrestrial network of meteorological stations in the Feldbach region of southeast Austria. It consists of 153 meteorological stations, constructed in an area of roughly 20 km x 15 km, forming a structured grid with a cell area for each station of about 2 km$^2$ (see Fig. 1). The orography and vegetation has no significant influence on the accuracy of the rain gauge measurements. Each station measures

meteorological quantities such as temperature, humidity and precipitation (see Kirchengast et al. (2014) for further details). In all stations a tipping bucket rain gauge instrument with a volume of 0.1 mm is used, however, only twelve contain a heating device, and are, therefore, able to reliably measure winter precipitation.





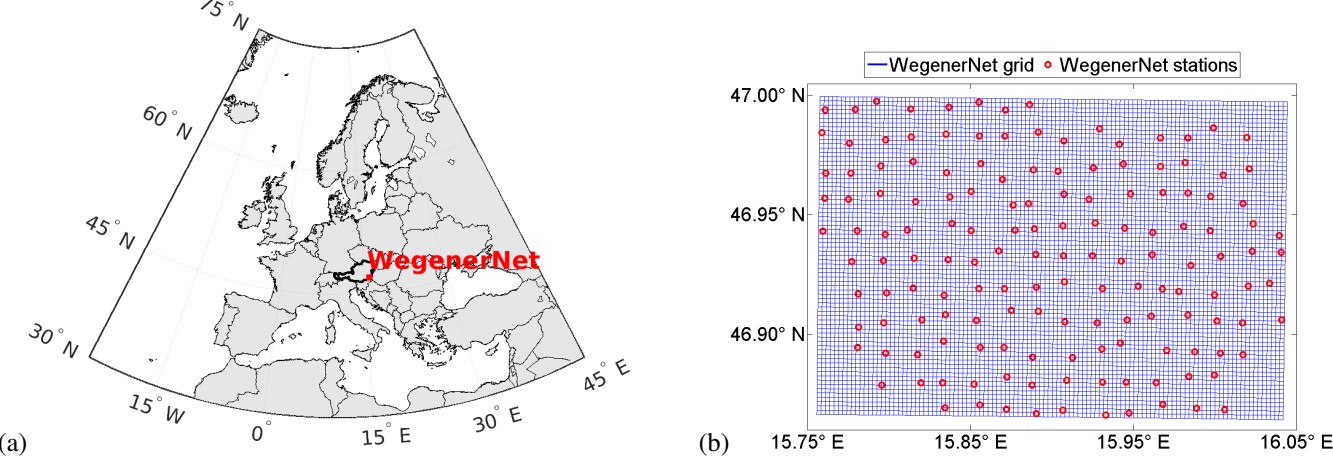

(a)   (b)

**Figure 1.**

(a) Location of the WegenerNet in Europe.

(b) Geometry of the WegenerNet; red circles indicate the meteorological stations, blue displays the 200 m grid; the area is roughly 20 km × 15 km. Measurements are taken in five minute accumulations.

Gauge data as ground reference are widely used in many existing validation studies (e.g. Amitai et al., 2015; Tan et al., 2017; O et al., 2017). Since satellite estimates provide only measurements at points in time and not accumulations, one prerequisite for evaluating with terrestrial gauge data is that the accumulations are as short as possible, but still providing high quality information. Even though the WegenerNet is of very small scale, it provides in its 5 minutes accumulations considerably more

5 and better of information, especially because of the high spatial resolution in the covered area. Additionally, the biases of the network are known and corrected (see O et al., 2018). Therefore, it is possible to evaluate every DPR footprint based on the information of 8-10 or even more stations. The drawback is the small number of fly-overs with actual precipitation, which allows only for an event-based analysis.

10 The paper is structured in four upcoming sections. Section 2 gives a description of the data, i.e. rain gauge measurements from the WegenerNet and quantitative precipitation estimates from the GPM-DPR. It is followed by Section 3, which introduces the focus and the methodology of validating GPM-DPR based on a small terrestrial network. Section 4 deals with the results of the validation and its limits. Finally, concluding remarks are pointed out in Sect. 5.





## 2 Data

### 2.1 WegenerNet

The WegenerNet is a network of high spatial resolution for weather and climate studies, located in the Feldbach region in southeast Austria. The region is characterised by moderate hilly landscape in the alpine foreland with altitudes between 260 m

to 600 m, and the valley of the river Raab. The network incorporates 153 weather stations in an area of about 300 $km^2$, employing tipping-bucket gauges to collect rainfall measurement data every 5 min. The general user data products are station time series, as well as a 200 m × 200 m gridded data set, calculated by applying an inverse-distance-weighted interpolated method. The data products are available online at the WegenerNet web portal (http://www.wegenernet.org/) within 2 h latency. The WegenerNet provides a robust ground reference, its data bias is well studied (O et al., 2018), and it has been used in

several other studies for satellite data validation (e.g. O et al., 2017; Kidd et al., 2017). Furthermore, the spatial uncertainties of rainfall over the WegenerNet were investigated by O and Foelsche (2018). More information on the WegenerNet and its data products can be found in Kirchengast et al. (2014). The tipping bucket rain gauge instruments collect water up to 0.1 mm, which is the minimum resolution of the WegenerNet precipitation measurements. Since only a small number of stations are heated, times with possibly solid precipitation are excluded from the evaluation (the meteorological winter, i.e. Dec. 1[st] to the

end of February). The data from the tipping-bucket gauges are accumulated to five minute samples, which is the basic data product of the WegenerNet. Thus, the basic data are a sum of measurements that refers to an interval in time (in contrast to the GPM-DPR estimates, which refer to a point in time). Further products include half hour samples, one hour samples, daily, monthly, seasonal and yearly data.

The high spatial resolution of the WegenerNet allows an investigation of each footprint based on multiple gauges. However,

due to its small extent and due to the low sampling frequency of GPM, the number of rainfall data samples is limited.

The minimum resolution, which is also roughly the precision, of the WegenerNet is twice as good as the GPM-DPR estimates' minimum resolution. However, there is no other precision (quality) information for the GPM-DPR estimates. Even though the uncertainties are only better by the factor of two, the WegenerNet provides accurate information, not only whether precipitation occurred but also on the amount of rain. Furthermore, it delivers reliable (clear precision information given) measurements

over all amounts of rain. The gridded precipitation and the station-wise precipitation reflect the same information (see Fig. A1 and A2 in the appendix). For the evaluation mainly the gridded gauge data were taken into account because of their higher resolution, and therefore, more information on extensive parts of the WegenerNet (such as the area covered by one footprint of the satellite's radar).

### 2.2 GPM-DPR

The GPM-DPR provides accurate rain rate estimations on a global level. The DPR radar instrument measures on two different channels (Ka-band at 35.5 GHz and Ku-band at 13.6 GHz) to obtain the three-dimensional structure of precipitation, including heavy (tropical) rainfall and light rainfall at mid-latitudes.

In general, the strength of the radar echoes is affected by attenuation due to precipitation. The amount of attenuation depends



on the frequency and the size of raindrops. The precipitation radar matches the transmission pulse timings and the radar beam position with the attenuated echo to estimate the size of a raindrop (see JAXA, 2017). The GPM-DPR uses differential attenuation between the Ku-band and the Ka-band frequencies and the Variable Pulse Repetition Frequency (VPRF) technique for high resolution rain rate estimates. The amount of rain is obtained in further processing based on various algorithms (see

Iguchi et al., 2015). The Ku-band precipitation radar (KuPR) has an observation swath of 245 km with 49 bins (Normal Scan - NS), each resulting in a circular footprint of 5.2 km diameter. The bins do not overlap. The KuPR is more sensitive to heavy and moderate rainfall. The KaPR on the other hand, has half of the swath size of KuPR with 120 km and 49 bins. 25 bins between KuPR and KaPR are overlapping, i.e. KaPR Matched Scan (MS). KaPR shall provide better information on light rainfall and snow. The second scan, that is provided for the KaPR is a high sensitivity scan with 24 bins, where the beams are interlaced

within the scan pattern of the matched beams (Iguchi et al., 2015). The range resolution is 250 m for KuPR and 250m/500m for KaPR. The Dual Frequency (DF) rain rate estimation combines Ka- and Ku-band information to the DPR product. It is available for NS, MS and HS. Where the swaths do not overlap KuPR data are provided in the DPR product (JAXA, 2017).

In order to acquire as many rain rate estimates as possible over the WegenerNet, the NS was taken into account for the Ku-band and the MS for the two other products. The minimum resolution is given by the documents (JAXA, 2017) with 0.2 mm for the

15 Ka-band and the merged product. The Ku-band estimates resolve a minimum 0.5 mm of rainfall. However, recent evaluations assign to the KuPR estimates the same quality as the Ka-band delivers (Tan et al., 2017; Hamada and Takayabu, 2016).

In contrast to the observations taken by the terrestrial stations, the radar measurements resemble only one point in time and are converted through algorithms (e.g. Iguchi et al., 2015) into a rain rate per hour. This, however, implies that the matching between observation time and location of the rainfall is crucial to the quality of the product. Even if the estimated rainfall is

20 correct, it needs to be located on the right spot at the right time.

## 2.3 Selected data

In order to compare the two kind of data sets, two requirements have to be met: First, the radar observations must cover the area of the terrestrial network and second, precipitation must occur during this short time interval. This reduces the possible events for the evaluation drastically.

To evaluate the GPM-DPR rain rate estimates with the WegenerNet, every event in the DPR data was sought after, where the satellite's swath of all three data types (NS for KuPR and MS for the other two) passes the WegenerNet and rain is detected in at least one of the three GPM-DPR products.

For the study period of four years, this yields to

- 426 visits of the GPM core satellite over the WegenerNet

30 - with > 4000 footprints.

- 24 events with rain detected and

- a sum of 253 footprints.



This gives an average of 10 footprints per fly-over. Each footprint covers approximately 8-12 stations.

If an event contains footprints in which the estimation delivers zero rain rate, this zero is taken into account as actual estimation and tested against the WegenerNet. Since the WegenerNet is of local scale, only up to ten times a month the satellite's ground track crosses the area. With an average of around 800 mm of rain per year, the region of Feldbach, where the WegenerNet is located, is not the most rainiest. Therefore, only 24 events were detected. Two events had to be excluded because of missing WegenerNet gridded gauge data. Table 1 lists the remaining events.

**Table 1.** Evaluated rainfall events. Note that the four highlighted events are analysed in detail.

| | date | avg. rain rate Ku-NS [mm h$^{-1}$] | avg. rain rate Ka-MS [mm h$^{-1}$] | avg. rain rate DPR-MS [mm h$^{-1}$] | avg. precip. WegenerNet [mm h$^{-1}$] |
|---|---|---|---|---|---|
| Event 1 | 2014-04-29 | 0.26 | 0.07 | 0.29 | 0.05 |
| Event 2 | 2014-05-17 | 0.32 | 0.00 | 0.17 | 0.13 |
| Event 3 | 2014-05-18 | 0.31 | 0.12 | 0.22 | 0.26 |
| Event 4 | 2014-06-13 | 0.00 | 0.00 | 0.00 | 0.00 |
| Event 5 | 2014-06-24 | 0.08 | 0.08 | 0.00 | 0.26 |
| Event 6 | 2014-07-02 | - | 0.00 | - | 0.00 |
| **Event 7** | **2014-07-10** | **0.28** | **0.12** | **0.29** | **0.30** |
| Event 8 | 2014-07-10 | 0.02 | 0.00 | 0.04 | 0.00 |
| **Event 9** | **2014-08-05** | **0.43** | **0.26** | **0.42** | **0.14** |
| Event 10 | 2014-08-13 | 0.11 | 0.04 | 0.15 | 0.00 |
| Event 11 | 2014-10-21 | 0.09 | 0.06 | 0.10 | 0.01 |
| **Event 12** | **2014-10-22** | **0.71** | **0.45** | **0.82** | **2.82** |
| Event 13 | 2015-06-15 | 0.06 | 0.05 | 0.06 | 0.00 |
| Event 14 | 2015-08-15 | 0.44 | 0.19 | 0.64 | 0.14 |
| Event 15 | 2015-10-10 | 0.40 | 0.07 | 0.43 | 0.93 |
| **Event 16** | **2016-05-02** | **2.02** | **2.37** | **2.16** | **2.43** |
| Event 17 | 2016-06-19 | 0.39 | 0.24 | 0.40 | 0.41 |
| Event 18 | 2016-06-27 | 0.46 | 0.22 | 0.48 | 0.09 |
| Event 19 | 2016-07-16 | 0.75 | 0.39 | 0.76 | 0.05 |
| Event 20 | 2017-05-15 | 0.21 | 0.06 | 0.27 | 0.15 |
| Event 21 | 2017-05-15 | 0.00 | 0.00 | 0.58 | 0.00 |
| Event 22 | 2017-08-28 | 0.00 | 0.00 | 0.00 | 0.00 |
| Average | - | 0.35 | 0.22 | 0.39 | 0.37 |





Within those events three events provide zero rain in WegenerNet and radar estimates (Events no. 4, 6, 22). These events are included because there was rain slightly outside of the WegenerNet observed by the satellite and the "zero rain" information gives an actual information that can be evaluated with the WegenerNet. Event no. 21 had to be omitted because of only one footprint partly covering the Southwest border of the WegenerNet in the MS. The NS (Ku-band) covered the whole network,

however, no rain was detected. Generally, the events show very light rain, only two of them have an average of more than 1 mm h$^{-1}$. The events in bold lettering are analysed in more detail; two with light rainfall (no. 7 and 9) and two with heavier rain (no. 12 and no. 16). Since GPM estimates are given in the unit of [mm h$^{-1}$], the WegenerNet data are converted from millimeter per 5 minutes to millimeter per hour.

The data can be easily visualised (see Fig. 2 for Event no. 16). Note that the circular footprint is distorted into an ellipse due to

10 the meridian convergence.

The GPM-DPR estimates provide one rain rate per footprint and the footprints do not overlap. In contrast to that, the WegenerNet has about 8 to 10 stations per footprint and one cell of the gridded rain gauges covers roughly an area of 200 m x 200 m, which sums up to around 500 grid box values per footprint. As one can see in Fig. 2, every footprint contains a large range of rainfall and a lot of variability. All of these shall be approximated by one single value the GPM-DPR delivers. For the

15 comparison in this study, the average of the gridded data within the footprint is taken as the most representative value for the WegenerNet and in a least squares sense it is the best estimation. The kind reader may keep in mind, that the GPM-DPR footprint rain rate estimates are treated as mean areal rainfall, thus also averaging intra-footprint rainfall. The DPR misses spatial information of highly variable rainfall events (inter-pixel rainfall variability) within a certain area (see bottom-right plot in Fig. 2). The fact, that the WegenerNet captures this inter-pixel rainfall variability and intra-footprint rainfall variability (bottom-left

graph of Fig. 2), makes it a robust ground reference. The most important statistical measures on the WegenerNet for 2$^{nd}$ of May, 2016 are provided in Table 2. During this event the areas outside of the footprints showed a similar behaviour as inside and as the whole network, but this is not necessarily the case for strong convective events.

**Table 2.** Statistical properties for the WegenerNet on 2$^{nd}$ of May, 2016 (Event 16)

|  | whole WegenerNet | inside footprints | outside footprints |
|---|---|---|---|
| Mean [mmh$^{-1}$] | 2.34 | 2.41 | 2.25 |
| Standard deviation [mmh$^{-1}$] | 1.12 | 1.14 | 1.08 |
| Normalised standard deviation [%] | 48 | 47 | 48 |

The large normalised standard deviation implies big variations within the whole area.



**Figure 2.** Measurements from the GPM-core satellite (Ku-NS, Ka-MS, DPR-MS) in and around the WegenerNet compared to WegenerNet grid data for the 2nd of May, 2016 including what is detected by the satellite and what is missed (lower graphs).



## 3   Methodology

The evaluation in this study is based on an interpretation of 22 events using graphical support (such as scatter plots) and mathematical tools: We adopt a correlation and a bias between the GPM-DPR and the WegenerNet and statistics based on a contingency table. The statistical items are the Proportion Correct (PC), the Frequency Bias Index (FBI), the Probability Of

Detection (POD), the False Alarm Ratio (FAR) and the Probability Of False Detection (POFD). The events are not interrelated, no time series analysis can be applied. Therefore, the interpretation is event-based.

The GPM-DPR delivers one rain rate value per footprint. These footprints are mapped to the WegenerNet gridded gauge data. All grid cells inside one circular footprint and the grid cells that are intersected by the footprint's border are accounted as the WegenerNet's equivalent to the footprint. As the WegenerNet observes multiple gauges per footprint, the arithmetic mean is

taken as most representative value. Even if the gauge observations within the footprint do not follow a Gaussian distribution, the mean value delivers a clear message about the regarded area.

In the following equations, $G$ denotes the GPM-DPR estimates and $W$ the WegenerNet. The correlation used is the Pearson's correlation coefficient, Eq. (1).

$$r = \frac{\sum_{i=1}^{n}(G_i - G_{\mathrm{mean}})(W_i - W_{\mathrm{mean}})}{\sqrt{\sum_{i=1}^{n}(G_i - G_{\mathrm{mean}})^2 \cdot \sum_{i=1}^{n}(W_i - W_{\mathrm{mean}})^2}} \tag{1}$$

The bias is calculated as the average of the deviation between the GPM-DPR estimates and the WegenerNet. It is not normalized with $W_{\mathrm{mean}}$ because of the light rainfall events, which would lead to a huge bias in case of a very small mean.

$$b = \frac{\sum_{i=1}^{n} G_i - W_i}{n} \tag{2}$$

For the contingency table, the estimates are compared to the ground reference and divided into four groups:

- Hits: Both systems provide precipitation information.

- False alarms: Only the GPM-DPR shows rain.

- Misses: The GPM-DPR does not deliver rain, whereas the WegenerNet does.

- Correct negatives: Both systems give no rain.

From the amount of these values, statistical items can be derived. The kind reader may note, that the number of correct negatives can easily take an effect on the results. Thus, the events to evaluate have to be carefully chosen.

The proportion correct (PC, Eq. 3) is the number of hits plus the number of correct negatives divided by the whole sample size ($n$), thus, providing an information about the number of correctly detected events. It ranges between 0 and 1 with a perfect score of one (i.e. 100% of the events show the same type of information between the GPM-DPR and the WegenerNet).

$$PC = \frac{N_{\mathrm{hits}} + N_{\mathrm{correct\ negatives}}}{n}. \tag{3}$$



The FBI (Eq. 4) gives an impression whether an over- or an underestimation occurs. It describes the ratio between the number of footprints that are detected by the GPM-DPR to feature precipitation and the number of footprints that show precipitation according to the WegenerNet. Its range is between 0 and $\infty$, with an perfect score of 1. An FBI larger than 1 means overestimation, <1 is an underestimation.

$$FBI = \frac{N_{\text{hits}} + N_{\text{false alarms}}}{N_{\text{hits}} + N_{\text{misses}}}. \tag{4}$$

The probability of detection (POD) is

$$POD = \frac{N_{\text{hits}}}{N_{\text{hits}} + N_{\text{misses}}}, \tag{5}$$

The POD ranges from 0 to 1, with one as all rain events detected correctly (no miss). It is only sensitive to missed events, this means, it can be (artificially) improved by overestimation, which leads to a reduction of misses. The increase of false alarms does not influence the POD. For a more sophisticated interpretation, the constraint of the GPM-DPR estimates being in an interval of $W_{\text{mean}} \pm W_{\text{std.-dev.}}$ is applied to the POD.

The FAR is taken into account to cross-check with the POD (Eq. 6).

$$FAR = \frac{N_{\text{false alarms}}}{N_{\text{hits}} + N_{\text{false alarms}}}. \tag{6}$$

Again the range is between 0 and 1, with 0 as perfect score, which means that there is no event, where the GPM-DPR sees rain and the WegenerNet does not. The FAR is not sensitive to misses, but to false alarms. Therefore, it can be improved by underestimation (reducing the possible amount of false alarms, but also increasing the possible amount of misses). POD and FAR together provide a robust information on the quality of the estimation.

Finally, the POFD is calculated as

$$POFD = \frac{N_{\text{false alarms}}}{N_{\text{false alarms}} + N_{\text{correct-negatives}}}. \tag{7}$$

It ranges between 0 and 1, where 0 is the perfect score. In contrast to the FAR the POFD takes the correct negatives into account, which may lead to a very low value. Underestimation improves the POFD.

## 4 Results

### 4.1 Evaluation of all rainfall events

The most basic evaluation is the series of footprints for all events (Fig. 3). Note that this is not a time series in the sense that any time series analysis can be applied (no uniformly spacing etc.), but still it is ordered in time. At a first glance, the GPM-DPR estimates resemble the WegenerNet quite well. Over- and underestimations as in Event no. 12, 16 and 19 happen, but the general structure can be seen in the satellite data. Even when the GPM-DPR estimates over- or underestimate the true precipitation, more than 70% of the GPM-DPR precipitation rates are within the range of the respective WegenerNet gauges





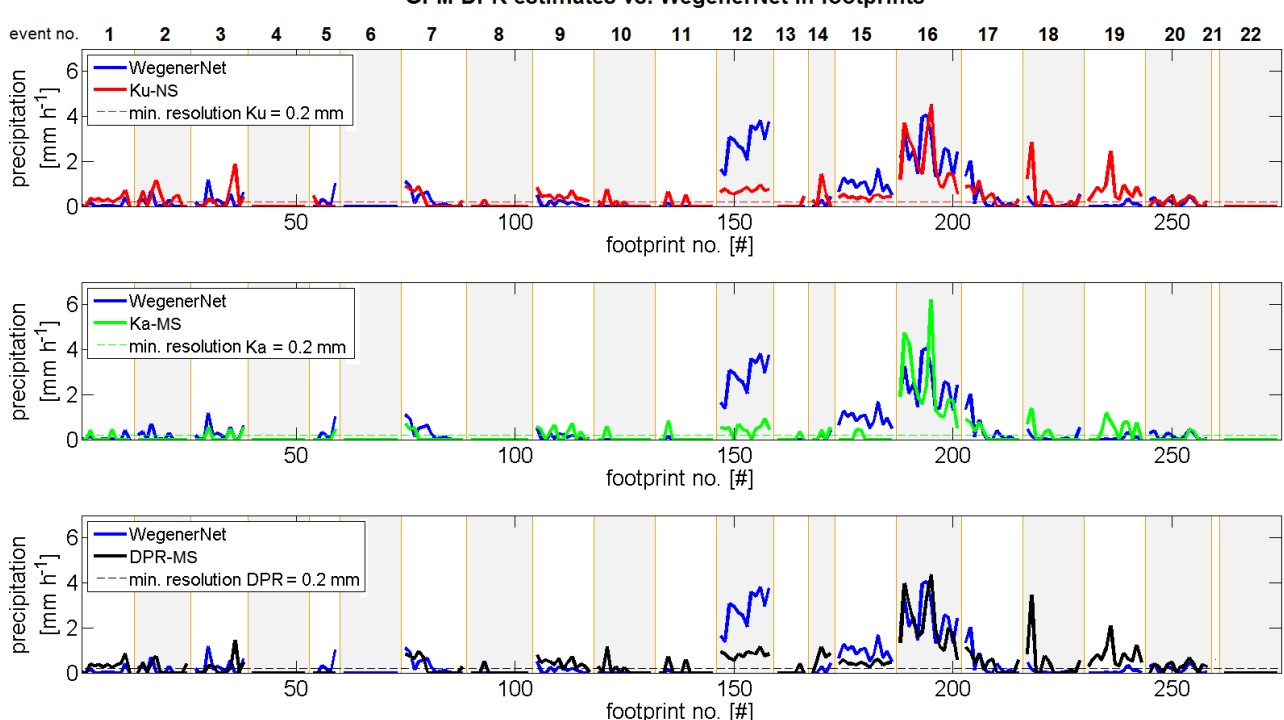

**Figure 3.** GPM-CO estimates over the WegenerNet (average of WegenerNet 5-min accumulation in the respective footprint) in the years 2014-2017, only events with precipitation detected in either Ku-NS, Ka-MS or DPR-MS were taken into account. The meteorological winter (Dec. 1st to the end of February) is excluded. Each box shows precipitation at a same time (one event per box) but different locations.

inside the footprint (for KaPR it is more than 90%) and almost 60% fulfill the tighter restriction of lying within one time the standard deviation around the average (73% for KaPR). See Fig. A3 in the appendix for a graphical representation.

Testing the GPM-DPR estimates directly against the WegenerNet precipitation leads to a scatter plot (Fig. 4). It shows, that the DPR tends to underestimate the WegenerNet gauges, as more estimations can be found below the diagonal. A point of interest

5   is when one system provides the information of zero rain and the other detects precipitation, which is basically a mis-detection if the WegenerNet does not see rainfall. However, since the mean of the gridded gauge data inside the footprint is taken into account, close to zero rainfall (rounded to zero) inside the footprint does not mean that no precipitation occurred. GPM-DPR probably observed a part of the WegenerNet grid boxes in its footprint area, where there was no rainfall, even though it rained in the other part of the WegenerNet grid boxes. This over-/underestimation of satellite precipitation estimates due to the subpixel-

10  scale rainfall variability was also found by O et al. (2017). Therefore, the detection of rain in the GPM-DPR cannot be treated as completely wrong without considering a certain interval around the mean (range inside the footprint and/or the standard deviation of the mean).

The most general statistics can be applied by setting up a contingency table and counting the number of hits, misses, false





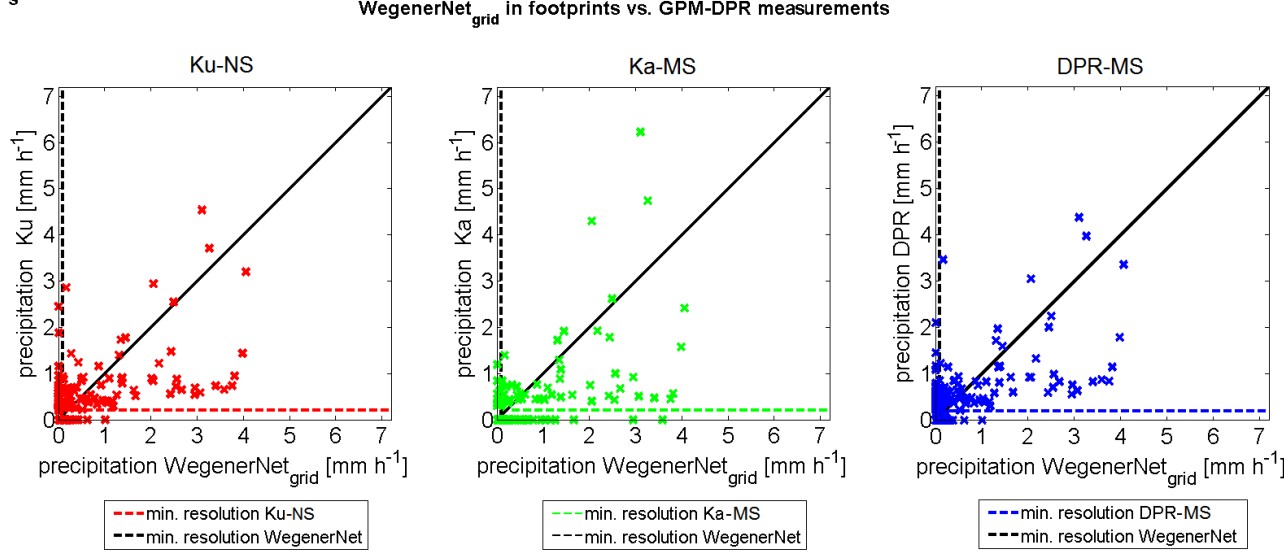

**Figure 4.** Average of GPM-DPR observations as a function of WegenerNet measurements inside the resp. footprints. The diagonal denotes the line where the satellite measures the same as the terrestrial network.

alarms and correct negatives (Table 3). The ground reference provides 126 times of precipitation inside a footprint and 127 times there was no rain detected. The Ku-NS product and the DPR-MS always score the same number, and are in lead when it comes to actual hits. The Ka-MS misses more than twice as much as the Ku-NS and hits only 2/3 of the KuPR, however it is more sensitive to delivering correct negatives. Furthermore, only five false alarms are given by KaPR. Even though the KaPR

5 scores as many hits as misses, it performs quite well due to its ability of correctly mirroring the WegenerNet's non precipitation events (few false alarms, lots of correct negatives). The PC, FBI, POD, FAR and POFD are items derived from the contingency table. These are given in Table 4. The PC is around 0.70 for all three products, which means that around 70% of all footprints correctly detect rain/no-rain. However, this number may be misleading, because the no-rain footprints get the same weight as the ones with rain. Therefore, a lot of no-rain events, correctly detected, increase the PC. In our case, the number of rainy

10 footprints is almost equal to the number of no-rain footprints, hence, it can be assumed that 70% is a good rate.

**Table 4.** Statistics derived from the contingency table for all events.

|      | Ku-NS | Ka-MS | DPR-MS |
|------|-------|-------|--------|
| PC   | 0.70  | 0.72  | 0.70   |
| FBI  | 1.10  | 0.52  | 1.10   |
| POD  | 0.75  | 0.48  | 0.75   |
| FAR  | 0.32  | 0.08  | 0.32   |
| POFD | 0.35  | 0.04  | 0.35   |



**Table 3.** Contingency table

| WegenerNet → | | yes | no | Σ |
|---|---|---|---|---|
| ↓ GPM-DPR | | | | |
| | | **hits** | **false alarms** | |
| | Ku-NS | 95 | 44 | 139 |
| yes | Ka-MS | 60 | 5 | 65 |
| | DPR-MS | 95 | 44 | 139 |
| | | **misses** | **correct negatives** | |
| | Ku-NS | 31 | 83 | 114 |
| no | Ka-MS | 66 | 122 | 88 |
| | DPR-MS | 31 | 83 | 114 |
| | Ku-NS | 126 | 127 | 253 |
| Σ | Ka-MS | 126 | 127 | 253 |
| | DPR-MS | 126 | 127 | 253 |

The FBI gives an impression about the under- or overestimation of the precipitation for a certain number of events. It does not take into account whether a single footprint was subject to mis-estimation or not. The correct negatives do not influence the FBI. The Ku-NS and the DPR-MS tend to overestimate (FBI = 1.10; more yes events in the KuPR than in the ground reference), whereas the KaPR underestimates the precipitation (FBI = 0.52).

The POD has a value of 0.75 for Ku and DPR-merged and 0.48 for the Ka-band. The better performance of the Ku-band results may be assigned to the fact that a lot of "correct negatives", i.e. both systems see zero rainfall, occur and these do not contribute to the POD. The probability of detection in each event, however, shows more discrepancies for the POD (see Table A1). According to Skofronick-Jackson et al. (2016) the POD on a global level is >64%, which shows that the GPM-DPR is suited to capture the precipitation that takes place over the WegenerNet. The good POD of Ku-NS and DPR-MS is supported by the FBI,

which shows a tendency to overestimation. Since the POD can be increased by overestimation (false alarms do not contribute to the POD), the FAR is considered as well. It shows that the rate is around 30% for Ku-NS and DPR-MS, whereas the KaPR has a FAR of only 8%, thus, delivering very few false alarms. The FAR can be improved by underestimation, which closes the circle to the FBI. The overestimation in Ku-NS and DPR-MS worsens the FAR, but enhances the POD, the underestimation through KaPR improves the FAR. Since the POD and the FAR do not consider the number of correct negatives, the POFD

sheds light on the probability of getting a false alarm compared to the correct negatives. Here, the KaPR and its tendency to underestimation (few false alarms), delivers best results.

Very low rainfall events are better detected by the Ka-band frequency, whereas heavier rainfall can be seen in Ku (as expected) and Ka. The dual frequency product is in between Ku-NS and Ka-MS with a tendency of giving more weight to Ku-band than Ka-band data. The events without rainfall over the WegenerNet are detected with a 100% score in all products.



Considering the constraint that the GPM-DPR estimates must be in an interval of ± standard deviation around the mean of the WegenerNet in the respective footprint, the POD for the series of events gives around 50% for the GPM-DPR estimates (see Table 5, bottom). This is an increase for the KaPR estimates, a decrease for the other two. The constraint adds a lot of misses to the statistics for Ku-NS and DPR-MS, while decreasing the number of hits. For the Ka-band estimates, this improves the statistical result slightly (PC = 0.74%, FBI = 0.56). For the Ku-NS and DPR-MS rain rate estimates the FBI decreases towards underestimation, while the FAR gets worse (less hits shrinks the FBI and increases the FAR). The PC drops to less than 60%. Since the POFD keeps its level, the number of false alarms is stable.

**Table 5.** Contingency table and statistics with the constraint that a hit is only scored when the GPM-DPR estimate is within the interval of $[W_{\mathrm{mean}} - W_{\mathrm{std.\text{-}dev.}}, W_{\mathrm{mean}} + W_{\mathrm{std.\text{-}dev.}}]$.

| | WegenerNet → | yes | no | Σ |
|---|---|---|---|---|
| ↓ GPM-DPR | | | | |
| | | **hits** | **false alarms** | |
| | Ku-NS | 65 | 44 | 109 |
| yes | Ka-MS | 65 | 5 | 70 |
| | DPR-MS | 67 | 44 | 111 |
| | | **misses** | **correct negatives** | |
| | Ku-NS | 61 | 83 | 144 |
| no | Ka-MS | 61 | 122 | 183 |
| | DPR-MS | 59 | 83 | 142 |
| | Ku-NS | 126 | 127 | 253 |
| Σ | Ka-MS | 126 | 127 | 253 |
| | DPR-MS | 126 | 127 | 253 |

| | Ku-NS | Ka-MS | DPR-MS |
|---|---|---|---|
| PC | 0.58 | 0.74 | 0.59 |
| FBI | 0.87 | 0.56 | 0.88 |
| POD | 0.52 | 0.52 | 0.53 |
| FAR | 0.40 | 0.07 | 0.40 |
| POFD | 0.35 | 0.04 | 0.35 |

The (counter-intuitive) improvement of the performance of Ka-band rain rate estimates when constraining, can be explained with very light rainfall events, where a Ka-band miss lies within the constraining range, and is consequently detected as a hit. In





a more balanced testing environment with more events of heavier rain, the KuPR would catch up. Notable is the DPR-MS estimation which is closer to the Ku-band estimates, even though both frequencies are combined. The influence of the constraint on the performance is very distinct in Event 12, where the simple POD gives values between 0.83 and 1 and the constrained version discards them.

The series of footprints of all events and the correlation and bias within one event is displayed in Fig. 5. There are no clear characteristics which GPM-DPR product catches the precipitation variations inside the WegenerNet better. Some events, e.g. no. 12, are not correlated. There is no event anti-correlated, which shows that the GPM-DPR estimates for one event are not completely shifted compared to the WegenerNet variations. The mean of the correlations over all footprints is $r = 0.42$ for
10  Ku-NS and $r = 0.47$ for Ka-MS and DPR-MS.

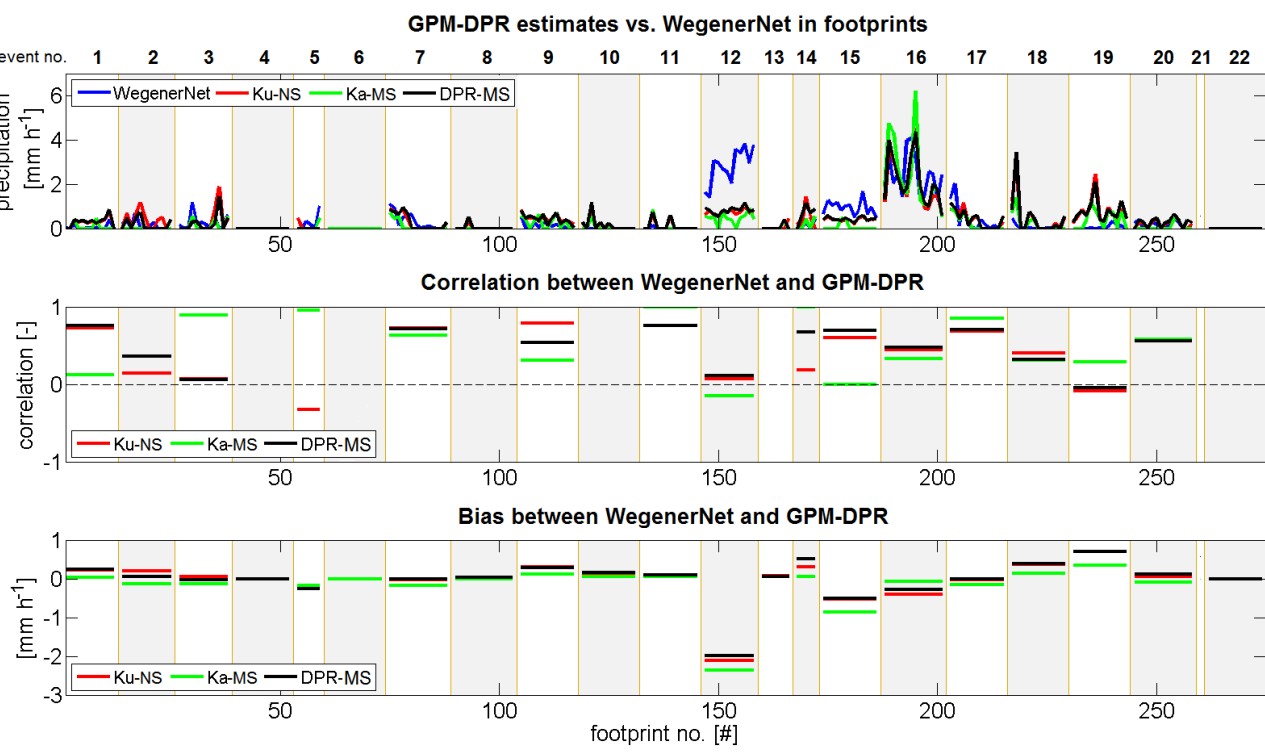

**Figure 5.** Correlation and bias between GPM estimates and the WegenerNet within all detected events. The topmost graph gives the precipitation of the events again.

Even though the bias can be high, there is no connection to the correlation. Perfect bias of zero occurs only at events without precipitation. Considering the bias and the correlation, one can derive, that the DPR-MS is closer to Ku-NS than to Ka-MS. Ka-band estimates have in general a lower bias than the others, this again may be explained with the high number of light rainfall events.





## 4.2 Analysis of example rainfall events

The example events where chosen based on the amount of rainfall in the WegenerNet. Two with light rain (no. 7 and 9) and two with moderate to heavy rain up to $7\,\mathrm{mm\,h^{-1}}$ in a certain footprint (no. 12 and 16). According to the rain type specified by the GPM-DPR data, all events are a stratiform phenomena, when Ku- or Ka-band data are considered, however dual frequency estimates state mostly convective rainfall. A look at WegenerNet reveals that the light rain events are more of a convective nature and the other two show stratiform behaviour. The events with light precipitation are in the hot season (July and August), the other ones in spring and autumn.

The first event to investigate is Event no. 7 on $10^{\mathrm{th}}$ of July, 2014, where the GPM-CO passed the WegenerNet at 11:40 and detected light precipitation. The chronological order of the footprints (each box depicts one footprint) and the estimated amount of precipitation is given in Fig. 6, the upper part showing the GPM-DPR estimates and the gauge station precipitation from the WegenerNet and the lower graph depicting the gridded gauge data instead of station data.

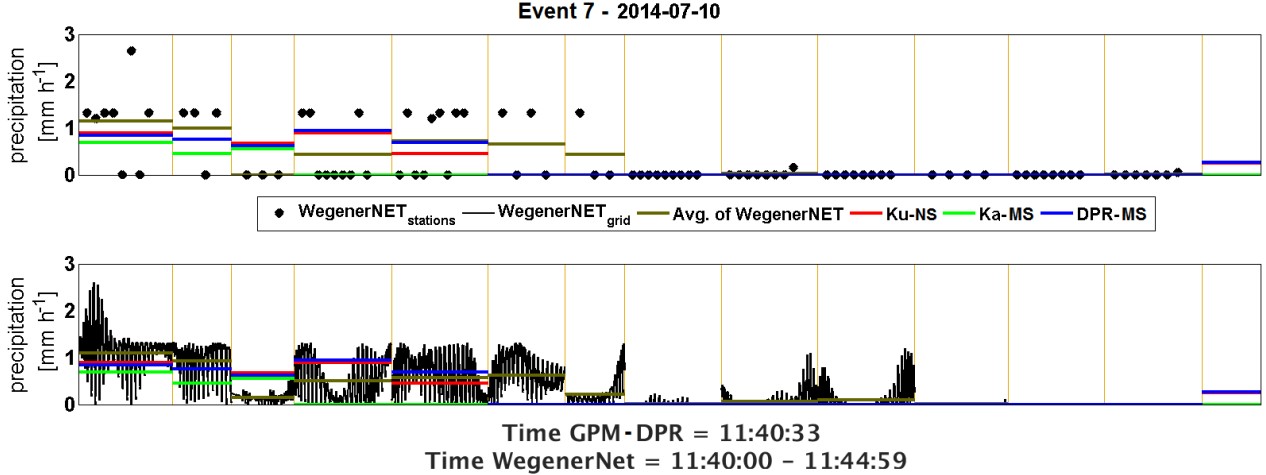

**Figure 6.** Series of footprints for Event 7 ($10^{\mathrm{th}}$ of July, 2014) compared to station and grid data.

More than half of the footprints do not feature any rain, which is correctly detected by the radar estimates. Within the other footprints, especially the ones in the middle of the graph, some information is added in the gridded gauge data compared to station alone because not only the gauges inside the footprint are used for the grid generation. All radar estimates are within the range of the respective WegenerNet footprint and close to the mean of the gauges. Thus a small bias is expected. The correlation between the terrestrial and the satellite data is close to one, which emphasises the quality of this event (see Event no. 7 in Fig. 5). The proportion between over- and underestimation is balanced inside the footprints. For the whole event the FBI supports a strong underestimation (see Table A1), strongly improved when adding the constraint (PC increases from 50% to 80%, FBI by the same amount). Interestingly, the Ka-band estimates, which should provide more accurate information in case of light precipitation, are not as accurate as the Ku-NS and the DPR-MS. The POD with constraint scores around 70% for





Ku-NS and DPR-MS estimates, a reasonably good value, since the GPM estimates have a small bias of less than 0.5 mm, but KaPR estimates are at only 55%. Without the constraint only 0.45 is reached, Ka-band is even worse, because of the zero rain estimation where the grid states very light rain. Indeed, not considering the gridded gauge data, but stations-only precipitation, would improve this result.

The second event (no. 9 on 8th of May, 2014) has even lighter rainfall than the first with a maximum of less than 1.5 mm. The series of footprints is given in Fig. 7. Nearly all stations inside the footprints indicate, that no precipitation was measured. The gridded WegenerNet precipitation however, shows some rainfall, which portends that a lot of information will be missed by the satellite.

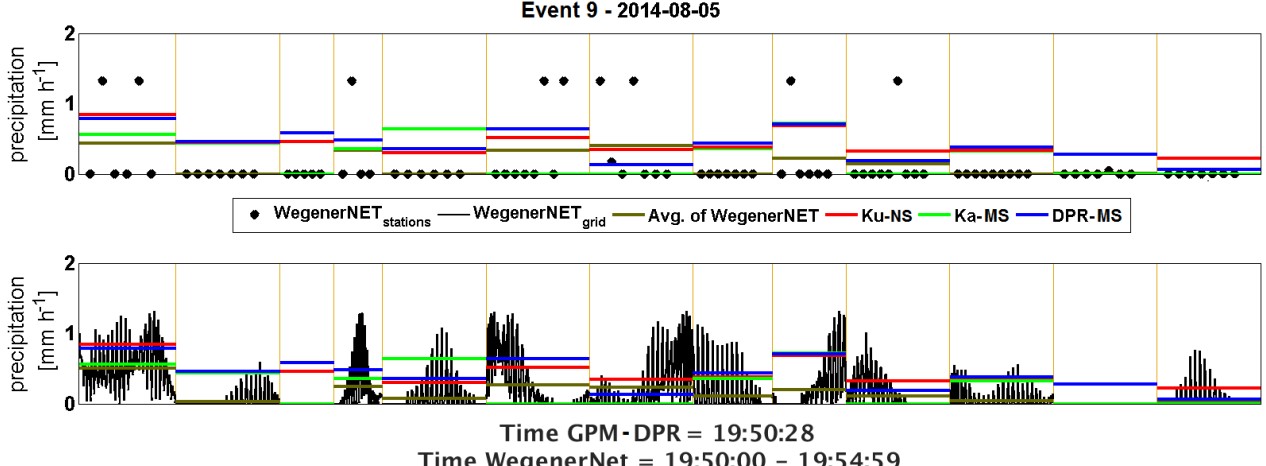

**Figure 7.** Series of footprints for Event 9 (8th of May, 2014) compared to station and grid data.

In contrast to Event no. 7, the GPM-DPR mirrors the gridded gauge data better than stations-only inside the footprint. The POD is at 1 for Ku-NS and DPR-MS, whereas the Ka-MS has a POD of only 0.64. The PC scores 0.85 for Ku-NS and DPR-MS and 0.69 for Ka-band estimates. The FBI suggests slight overestimation for the event, however the FAR is very low (between 0 and 0.15). When adding the constraint of the range, the Ka-band scores highest (0.55 compared to 0.27 for Ku-band and 0.36 for DPR-MS). The FBI and FAR now show an underestimation in Ku-band, Ka-band and DPR-MS estimates (FBI around

0.5, FAR between 0 (KaPR) and 40% (KuPR)), where Ka-band scores slightly better. This can be explained by the fact, that footprints out of range are treated as miss, which inverses the FBI's message. The Ka-band seems to be least sensitive to over-/underestimations. The correlation (Fig. 5) shows high discrepancies between the Ku-band, Ka-band and DPR-MS, with the Ka-MS being least accurate. A closer look at the event in the WegenerNet (Fig. A6, lag +0 min), points out that the event is hard to detect, since the precipitation occurs quite spotty over the whole area. Therefore, the satellite misses a lot of information

that can only be provided by the gridded gauge data.



For Event no. 12, represented in Fig. 8, all three GPM-DPR products show relatively poor performance. Every footprint is heavily underestimated, and the radar estimates are within the range only in three out of twelve footprints.

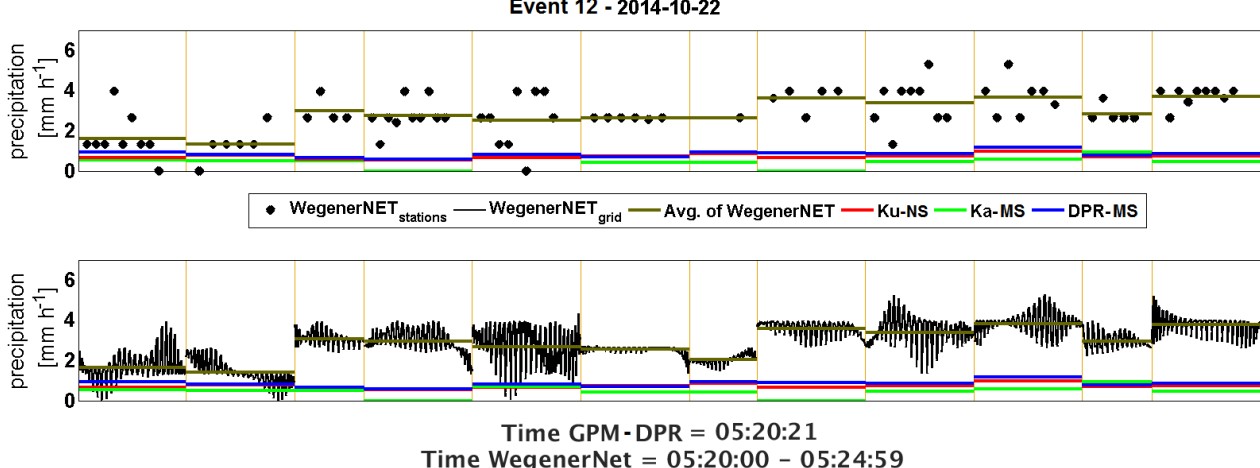

**Figure 8.** Series of footprints for Event 12 (22$^{nd}$ of October, 2014) compared to station and grid data.

The event contains uniformly distributed moderate rainfall (see figure A7 in the appendix), with hardly any station observing no-rain. The PC, FBI and POD are very high (between 83% and 100% without constraint), thus the fact, that it was rainy, is detected. However, the constrained POD discards all precipitation estimates. Ka-band information performs worst in terms of bias and correlation.

The opposite happens in Event no. 16, (2$^{nd}$ of May, 2016, Fig. 9), featuring moderate to heavy precipitation, up to 7 mm h$^{-1}$.

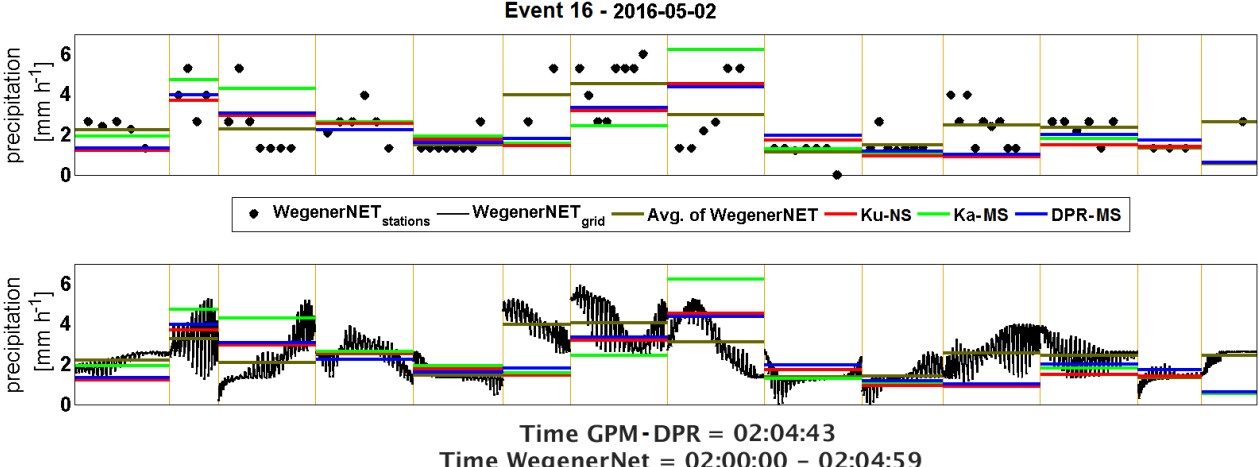

**Figure 9.** Series of footprints for Event 16 (2$^{nd}$ of May, 2016) compared to station and grid data.





The events shows a lot of variability (see Fig. 11 for 65 min of the rainfall) and is at a first glance almost perfectly mirrored by the radar estimates. The PC, FBI and POD is 100% for all products, the FAR is zero. The bias itself is quite low (highest for Ka-MS) and the correlation is close to 0.5. However, since there are no large scale variations within many footprints, the constrained POD drops to less than 0.6 for Ku-NS and DPR-MS. For Ka-MS, which is even out of the range of five footprints,

it is less than 0.3. The FBI states an underestimation.

Remote sensing data may show a time lag error between rain drops from clouds and the surface rainfall. In order to investigate this effect, whether the GPM-DPR estimation was matched to the correct point in time for the rainfall event, a lag of $\pm 30$ min is applied and the correlation and the bias are determined. This is displayed in Fig. 10.

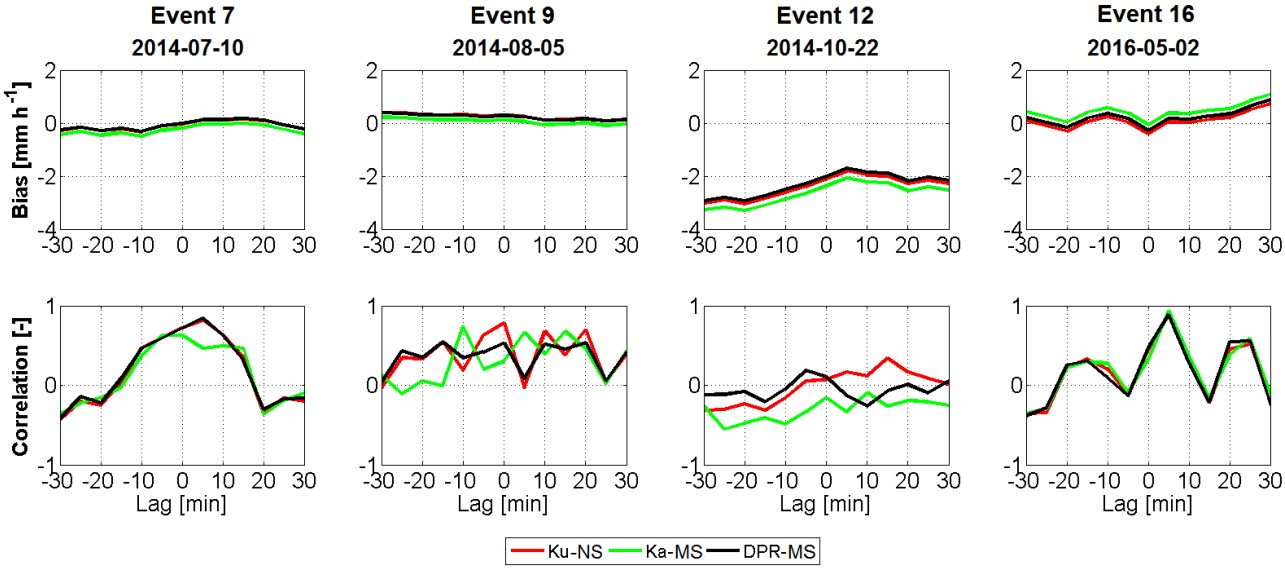

**Figure 10.** Bias and correlation for a lag of $\pm 30$ minutes for the example rainfall events.

For a perfectly matching event, the bias should be as close as possible to zero and the correlation should tend towards one, both

occur for example for the Ku-band at a lag of +5 min in Event no. 16. But one cannot conclude, that the lag with the smallest bias shows also the highest correlation.

Event 7 and 9 have a very small bias due to their light and non-extensive rainfall. When shifting the WegenerNet $\pm 30$ min the bias stays at the same level, with its closest point to zero at a lag of +0 min. From that point of view, a perfect matching in time is achieved. In case of Event 9 the correlation shows a lot of variation between the three products, still the Ku-band has

its peak at +0 min. Since there is a lot of information missed by the satellite, this event is hard to detect. In case of Event 7, the GPM-DPR footprints got exactly the characteristics of the WegenerNet at a lag of +5 min. A look at the lag +0 min and +5 min (Fig. A5) shows the similarities between the two lags. The fast decreasing correlation around the peak implies a fast moving precipitation event.

Event no. 12 is almost not correlated for the whole lag and also the bias is very high with only underestimated precipitation




rates.

For Event 16 a clear peak in the correlation is at a lag of +5 min, whereas -5 min and +15 min is almost not correlated. Thus, it was a quite fast moving rainfall, which underlines the importance of a correct tagging of rain rate estimates in time. The peak at +5 min lag can be explained in two ways. First the GPM-DPR measurements, which refer to a point in time, are taken at the very end of the WegenerNet interval. And second, the main characteristics of the whole WegenerNet which the satellite maps, did not shift from +0 min to +5 min, although the grid changed obviously (Fig. 11). One may compare with Fig. 2 to see which information is missed by the satellite.

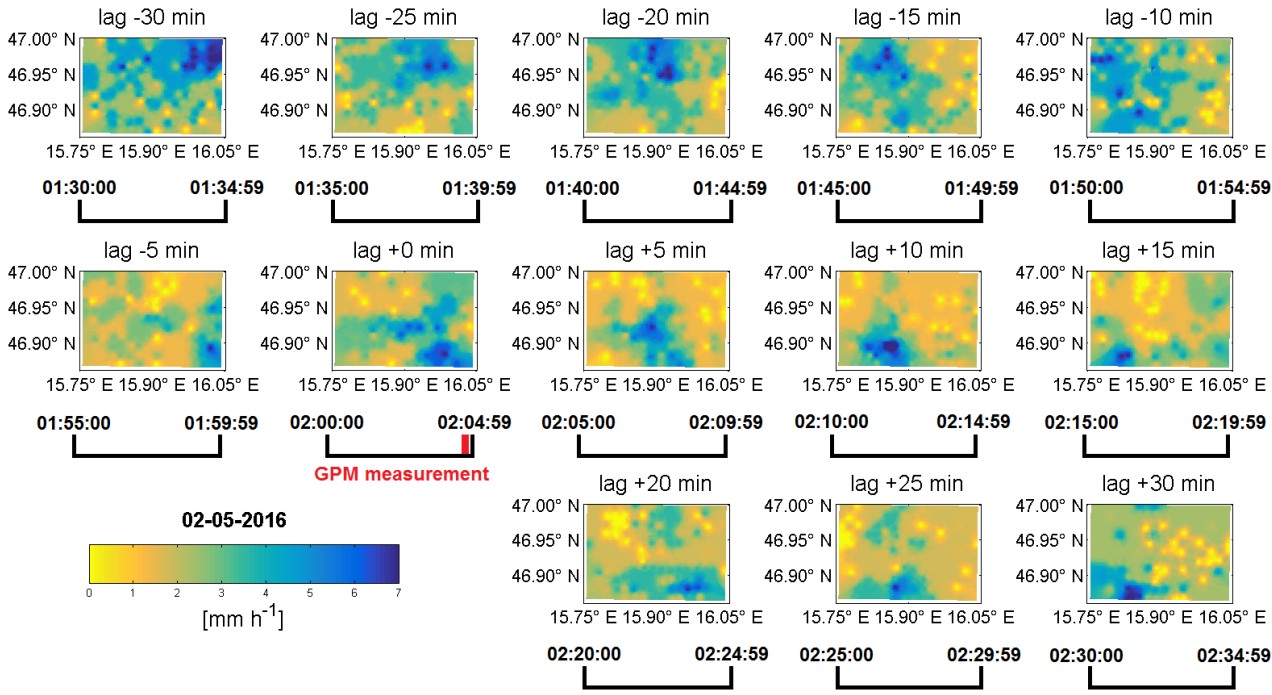

**Figure 11.** Evolution of Event 16 ($2^{nd}$ of May, 2016) with a lag of $\pm 30$ minutes.

## 5  Conclusions

In this study the radar estimates on both frequencies of the GPM-CO were evaluated using gauge measurement data from the WegenerNet network in southeastern Austria for the period of March–November of 2014 until 2017. The dense network provides the opportunity of evaluating the radar estimates not only amount-based but also on a level whether the satellite can observe small scale variability of rainfall events. Our results show that the evaluation using gridded gauge data provides more information than stations only. This plus of information helps to evaluate the GPM-DPR estimates and supports the quality of the satellites measurements in most cases. However, exceptions to that assumption can be found as well, especially in the case of light and spotty rainfall. One cannot infer the quality of the estimates from the amount of rainfall. In this study Ka-band





estimates perform best, probably due to the higher number of light rain events and their tendency of correctly mirroring zero rainfall. Consequently, Ka-band estimates tend to underestimation. Ku-band estimates and the dual frequency product perform vice versa. Considering the inter-footprint variability, all three products tend towards an under-representation of the precipitation. The correlation peak between the GPM-DPR and the WegenerNet is shifted in some events, however, this could be

explained by the distribution of the rainfall event and which parts of the network's area the DPR cannot see. The probability of detection is over 70% for Ku-band-NS and DPR-MS, but only 50% for Ka-band-MS. When the constraint of an interval of ±standard deviation is added, the POD for Ku-NS and DPR-MS estimates drops to less the 60%, the Ka-band keeps its level, showing that the Ku-NS and DPR-MS estimates tend to capture the precipitation, but largely biased.

The intra-event variations are captured from the satellite without clear characteristics, some events are resembled with a corre-

lation close to one and some are almost not correlated. But, there is no event completely anti-correlated. Any systematic shifts (by moving the WegenerNet ±30 min) could be explained by the gaps between the footprints.

Further studies on the GPM-DPR data and the WegenerNet, especially ones dealing with the detection of variations in small areas, would have to deal with the HS swath of the KaPR and DPR, because the amount of uncovered area inside the WegenerNet is drastically reduced.

*Data availability.* WegenerNet data are available at the WegenerNet data portal http://www.wegenernet.org/ in NetCDF format.
GPM-CO radar data sets are available at the PMM server http://pmm.nasa.gov/data-access/ delivered in HDF5 format.
Ku-band: doi:10.5067/GPM/DPR/KU/2A/05
Ka-band: doi:10.5067/GPM/DPR/KA/2A/05
DPR: doi:10.5067/GPM/DPR/GPM/2A/05





## Appendix: Appendix A

Table A1 lists the statistical items from the derived from the contingency table for each event. In Fig. A1 and Fig. A2 a graphical representation of the WegenerNet gauge observations is given, that compares the station-wise mean to the grid mean inside the GPM-DPR footprints. They contain the same information. Figure A3 depicts the series of footprints containing the range of the grid in each footprint (inter footprint variability, marked by the blue line), the standard deviation of the WegenerNet in each footprint and the difference to the respective GPM-DPR estimation. Ideally, the difference should be less than the standard deviation, which itself is less than the range. Figure A4, A5 and A6 show the evolution of the rainfall events, that were analysed in detail.

*Competing interests.* The authors declare that they have no conflict of interest.

*Acknowledgements.* The study was funded by Austrian Science Fund (FWF) under research grant W 1256-G15 (Doctoral Programme Climate Change Uncertainties, Thresholds and Coping Strategies).





**Table A1.** Statistics derived from the contingency table for each event.

| | PC | | | FBI | | | POD | | | FAR | | | POFD | | |
|---|---|---|---|---|---|---|---|---|---|---|---|---|---|---|---|
| | Ku-NS | Ka-MS | DPR-MS | Ku-NS | Ka-MS | DPR-MS | Ku-NS | Ka-MS | DPR-MS | Ku-NS | Ka-MS | DPR-MS | Ku-NS | Ka-MS | DPR-MS |
| Event 1 | 0.50 | 0.58 | 0.50 | 1.29 | 0.29 | 1.29 | 0.71 | 0.29 | 0.71 | 0.44 | 0 | 0.44 | 0.80 | 0 | 0.80 |
| Event 2 | 0.58 | 0.58 | 0.58 | 1.20 | 0 | 0.80 | 0.60 | 0 | 0.40 | 0.50 | - | 0.50 | 0.43 | 0 | 0.29 |
| Event 3 | 0.50 | 0.33 | 0.50 | 0.45 | 0.27 | 0.45 | 0.45 | 0.27 | 0.45 | 0 | 0 | 0 | 0 | 0 | 0 |
| Event 4 | 1 | 1 | 1 | - | - | - | - | - | - | - | - | - | 0 | 0 | 0 |
| Event 5 | 0.17 | 0.50 | 0.33 | 0.25 | 0.25 | 0 | 0 | 0.25 | 0 | 1 | 0 | - | 0.50 | 0 | 0 |
| Event 6 | 0 | 1 | 0 | Inf | - | Inf | - | - | - | 1 | - | 1 | 1 | 0 | 1 |
| **Event 7** | 0.50 | 0.43 | 0.50 | 0.55 | 0.27 | 0.55 | 0.45 | 0.27 | 0.45 | 0.17 | 0 | 0.17 | 0.33 | 0 | 0.33 |
| Event 8 | 0.93 | 1 | 0.93 | Inf | - | Inf | - | - | - | 1 | - | 1 | 0.07 | 0 | 0.07 |
| **Event 9** | 0.85 | 0.69 | 0.85 | 1.18 | 0.64 | 1.18 | 1 | 0.64 | 1 | 0.15 | 0 | 0.15 | 1 | 0 | 1 |
| Event 10 | 0.69 | 0.92 | 0.69 | Inf | Inf | Inf | - | - | - | 1 | 1 | 1 | 0.31 | 0.08 | 0.30 |
| Event 11 | 0.92 | 1 | 0.92 | 2 | 1 | 2 | 1 | 1 | 1 | 0.50 | 0 | 0.50 | 0.08 | 0 | 0.08 |
| **Event 12** | 1 | 0.83 | 1 | 1 | 0.83 | 1 | 1 | 0.83 | 1 | 0 | 0 | 0 | - | - | - |
| Event 13 | 0.86 | 0.86 | 0.86 | Inf | Inf | Inf | - | - | - | 1 | 1 | 1 | 0.14 | 0.14 | 0.14 |
| Event 14 | 0.40 | 1 | 0.60 | 1.50 | 1 | 2 | 0.50 | 1 | 1 | 0.67 | 0 | 0.50 | 0.67 | 0 | 0.67 |
| Event 15 | 1 | 0.15 | 1 | 1 | 0.15 | 1 | 1 | 0.15 | 1 | 0 | 0 | 0 | - | - | - |
| **Event 16** | 1 | 1 | 1 | 1 | 1 | 1 | 1 | 1 | 1 | 0 | 0 | 0 | - | - | - |
| Event 17 | 0.62 | 0.69 | 0.62 | 0.89 | 0.56 | 0.89 | 0.67 | 0.56 | 0.67 | 0.25 | 0 | 0.25 | 0.50 | 0 | 0.50 |
| Event 18 | 0.62 | 0.62 | 0.62 | 0.86 | 0.57 | 0.86 | 0.57 | 0.43 | 0.57 | 0.33 | 0.25 | 0.33 | 0.33 | 0.17 | 0.33 |
| Event 19 | 0.54 | 0.69 | 0.54 | 1.86 | 1 | 1.86 | 1 | 0.71 | 1 | 0.46 | 0.29 | 0.46 | 1 | 0.33 | 1 |
| Event 20 | 0.64 | 0.29 | 0.57 | 0.75 | 0.17 | 0.83 | 0.67 | 0.17 | 0.67 | 0.11 | 0 | 0.20 | 0.50 | 0 | 1 |
| Event 21 | 1 | 1 | 0 | - | - | Inf | - | - | - | - | - | 1 | 0 | 0 | 1 |
| Event 22 | 1 | 1 | 1 | - | - | - | - | - | - | - | - | - | 0 | 0 | 0 |





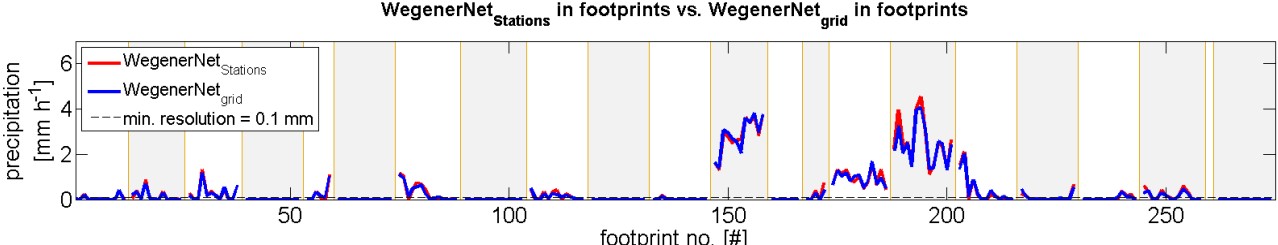

**Figure A1.** WegenerNet station data compared to the grid data in the footprints (average respectively).

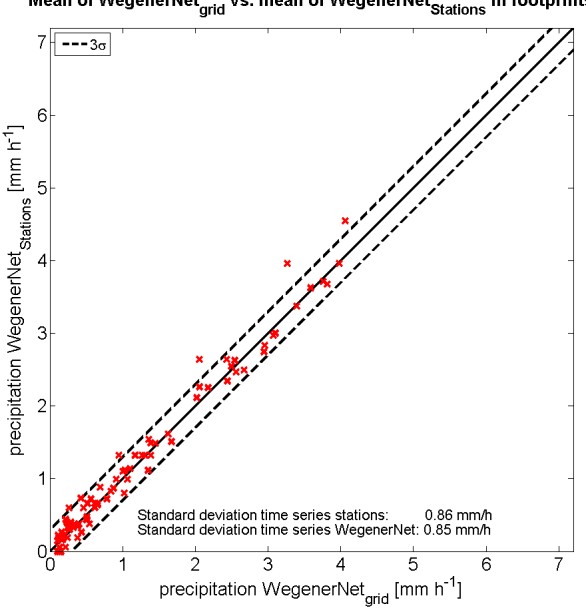

**Figure A2.** WegenerNet station observations as a function of the WegenerNet grid data. The dashed line denotes three times the standard deviation of the measurements.




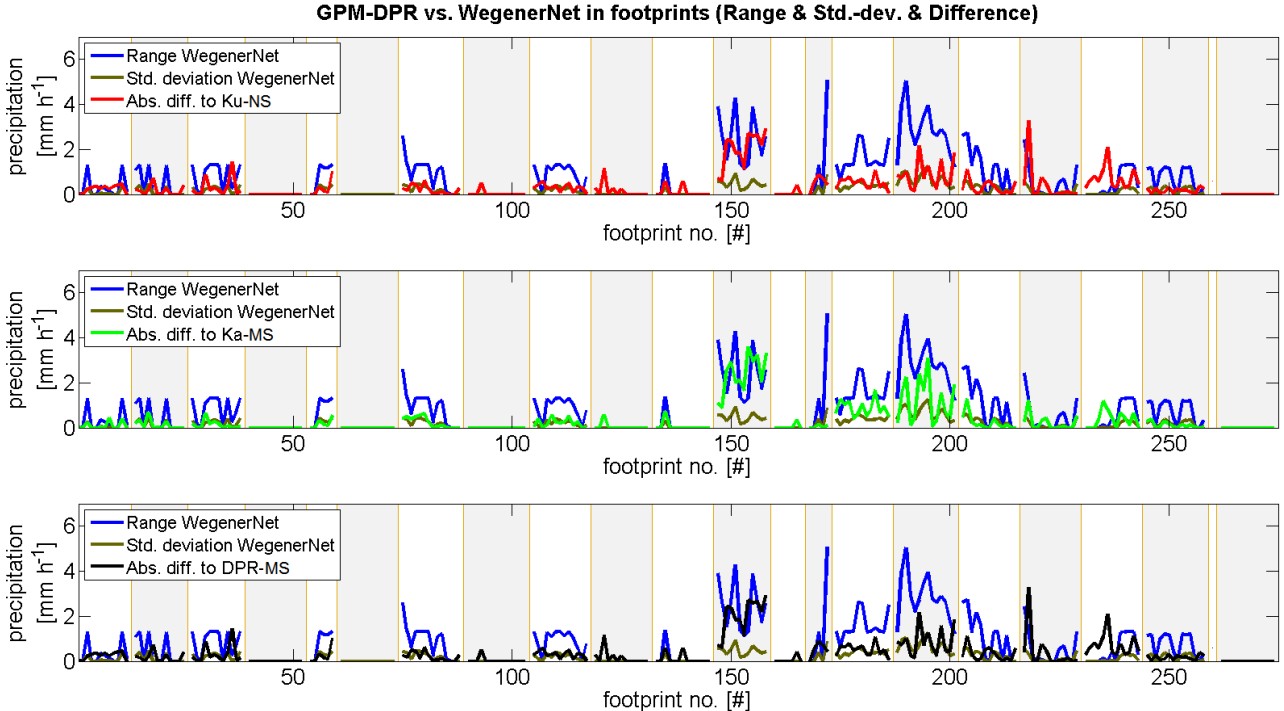

**Figure A3.** Range and standard deviation of the WegenerNet in the respective footprints and absolute difference between the GPM-DPR measurements and the average of the WegenerNet in the footprints.





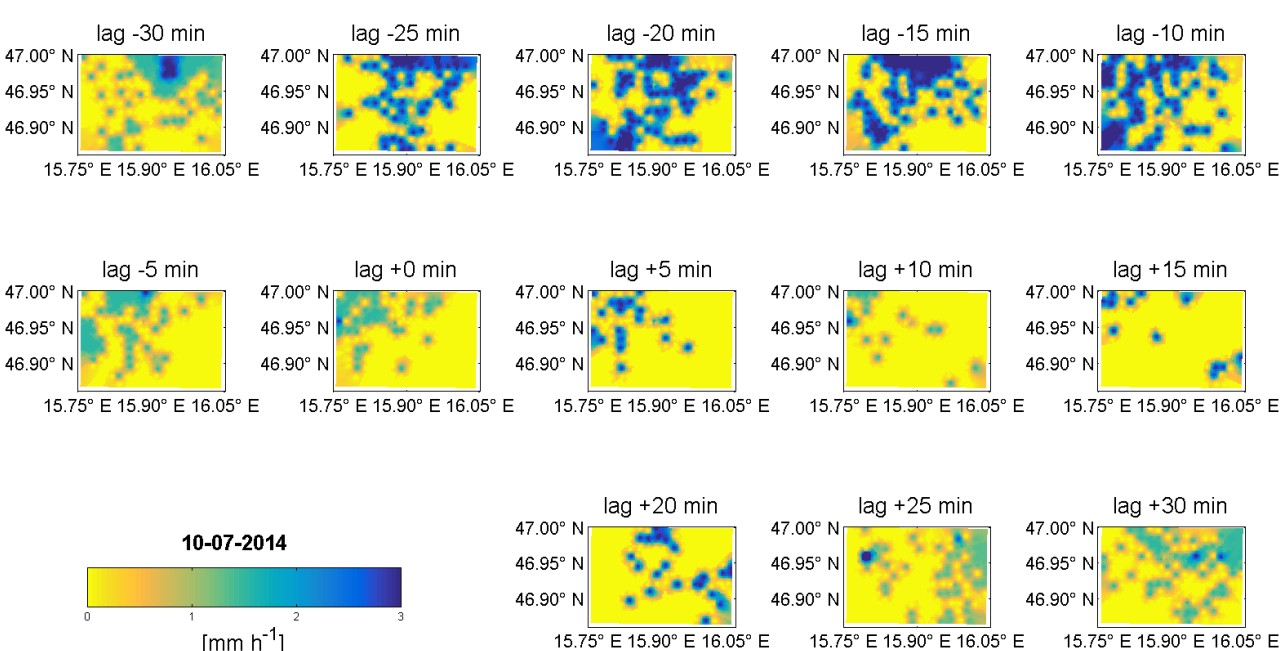

**Figure A4.** Evolution of Event 7 (10$^{th}$ of July, 2014) with a lag of $\pm$30 minutes.



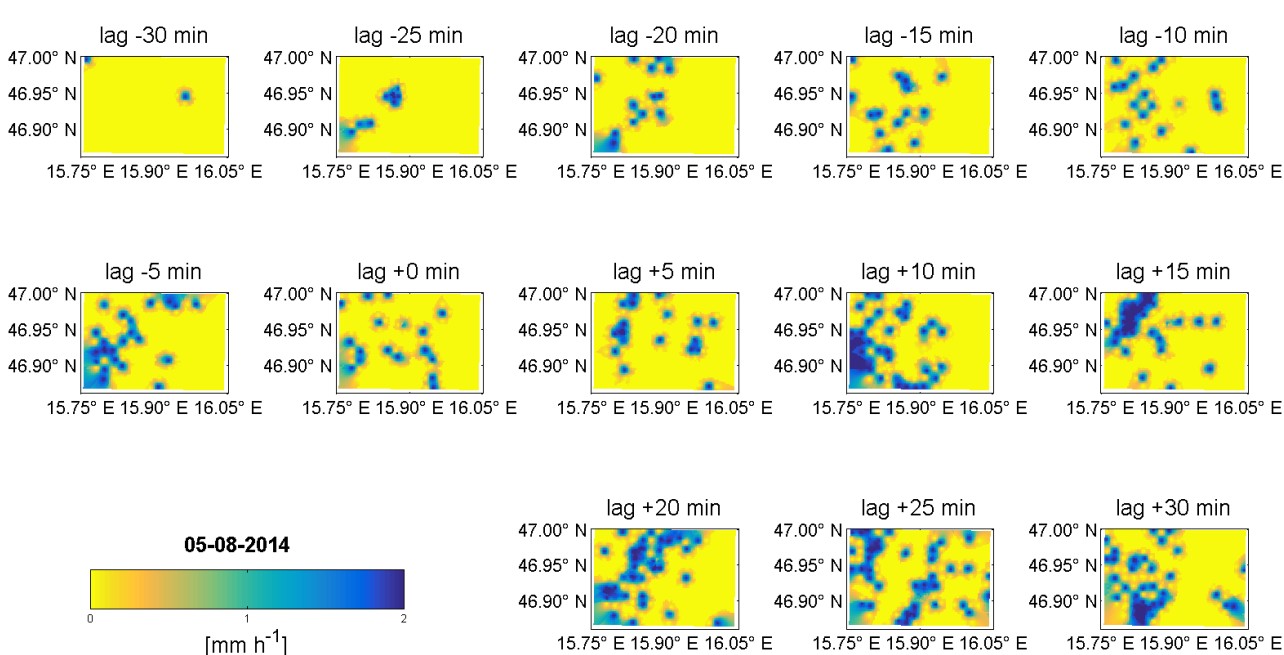

**Figure A5.** Evolution of Event 9 (8[th] of May, 2014) with a lag of ±30 minutes.





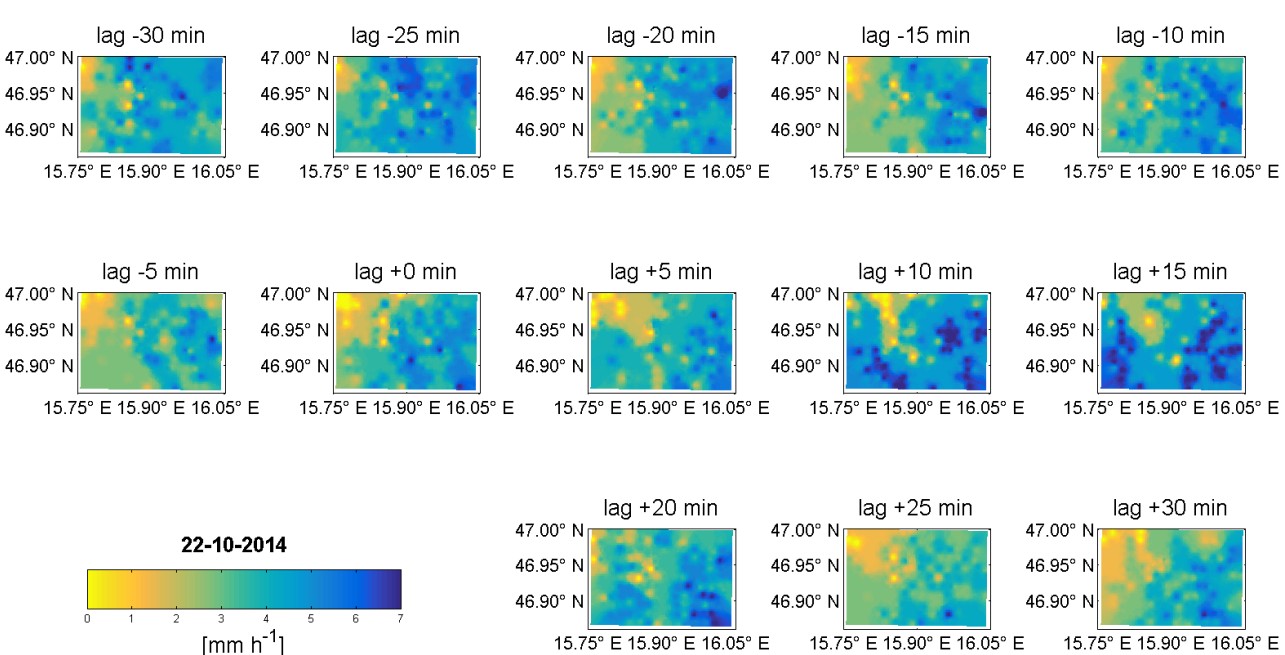

**Figure A6.** Evolution of Event 12 (22$^{nd}$ of October, 2014) with a lag of ±30 minutes.





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
