# Peer review of "Evaluation of GPM-DPR precipitation estimates with WegenerNet gauge data"

_Atmospheric Measurement Techniques, 2018_

## Referee Comment (RC1) · Anonymous Referee #1 · 19 Dec 2018

The paper "Evaluation of GPM-DPR precipitation estimates with WegenerNet gauge data", by Lasser and co-workers, presents a validation study of DPR precipitation products after comparison with rainfall rate measured by a very dense and well maintained raingauge network in the Alpine region. 22 precipitation events are considered and a number of statistical indicators are computed to quantify the accuracy of the DPR products.

The paper is fairly well written, and the topic falls within the scope of the journal. I feel, however, that experiment is not well designed for two main reasons.

First, the cumulation time selected for the raingauges is too short. If I well understood, the sensitivity of the TB is 0.1 mm, so that any measure X has to be intended as X∓0.1 mm. By considering such short time, most of the time (given the low rainrates

measured in the area) the data should be (looking at figures 6 to 9) for a single gauge 0.1 mm, resulting in a value of 1.2ïĆś1.2mm/h, that means a relative error of 100% for the most frequent value. Tokay et al., (2003, J. Atmos. Oceanic Technol., 20, 1460-1477) shows that for very light rain amount, the correlation between co-located tipping bucket raingauges can be very low. I suggest to use longer raingauge cumulation intervals (see also Porcù et al., 2014, Atmospheric Research) and to discuss the error associated to the ground measure.

The second issue is on the use of binary indicators to assess the quality of the DPR estimates. Two indicators (PC and POFD) includes the number of correct negatives as input. This number should not be considered in the evaluation, since can be arbitrarily larger or smaller by changing the selection of cases, and thus the results are not general. Moreover, I suggest to use other indicators (ETS, HSS, HK) to synthetize POD and FAR information.

More specific comments follow.

Page 2, Lines 18-19. The imager is only one (GPM Microwave Imager) and it is designed to provide a radiometric standard for the other radiometers of the GPM constellation.

Page 3, Line 1. There are two recent papers (Speirs et al, 2017; Petracca et al., 2018) performing a similar analysis in the same region (Switzerland and Italy): they are reported in the reference list but not mentioned in the paper. Also the important paper Seto and Iguchi (2014) is in the reference but not mentioned in the text. The Authors should carefully read these papers and discuss their results.

Page 4, Lines 12-15. No solid precipitation in summer (i.e. hail)?

Page 4. Line 27. Inverse distance interpolation does not add information to the gridded data, since the only information is in the raingauge measurements. The increased resolution can be good or bad, depending on how the precipitation pattern agrees with

the inverse distance model.

Page 5, Line 1. Frozen hydrometeors attenuate radiation as well.

Page 5, Lines 19-20. It has to be considered that the radar measures precipitation in a volume at a given altitude (near surface bin, I guess), and it takes time to the raindrops to reach the ground. In case the Authors claim for a very precise temporal matching, this issue should be discussed. It should also be mentioned if the "near surface bin" value is used to evaluate the vertical distance between the DPR estimate and the ground.

Page 7, Line 12. It is "8 to 10" or 8-12 (Page 6, Line 1) stations for footprint?

Page 9, Lines 15-16. Bias and Normalized Bias give different information and should be computed both. The fact that the NB gives huge values simply tells that the error is much larger than the measurement, this is often the case when a too short cumulation time is used (5 min).

Page 9, Lines 19-22. To build any contingency table the threshold to define rain and no-rain sample has to be carefully defined and reported here.

Page 9, Lines 23-24. As mentioned, the direct use of correct negatives should be avoided in any validation study (see Nurmi, 2003, Recommendations on the verification of local weather forecasts. ECMWF Technical Memoranda. Technical Memorandum No. 430 for reference). What is the precise meaning of the "careful choice" of the events? For this reason the indicator PC and POFD should be removed by the analysis.

Page 10, Line 21. A key indicator is the Equitable Threat Score (see Nurmi, 2003) that summarizes both POD and FAR, and gives the skill with respect to the random assignment of rainy footprint.

Page 10, Line 25-26. This sentence is useless and should be cancelled.

Page 10, Line 27. What is the "general structure"?

Page 10, Lines 28. What does it mean "70% of the GPM-DPR precipitation rates are within the range of the respective WegenerNet gauges"? What is the "range of WegenerNet gauges"?

Page 11, Line 7. What does it mean "close to zero"? How are the numbers rounded? How many significant digits are considered?

Page 12, Lines 8-10. Please avoid misleading numbers and cancel PC and POFD from the analysis.

Page 13, Lines 1-4. FBI gives a measure of the under-/overestimation of the wet area, not of precipitation.

Page 14, Lines 1-end. I'm not sure this analysis is well designed. If I understood well, hit is when the DPR products falls within +- 1 standard deviation of the corresponding gauges value, misses is if the DPR is lower than the gauges – 1 standard deviation, and false alarm if the DPR is higher than the gauges+1standard deviation value. Who are the correct negatives? How large is the standard deviation (roughly)? The main problem I see is that the distribution of rainrates is strongly asymmetric (power law) so that the standard deviation is asymmetric with respect to the mean value. In case of very light rainrate the value gauge-1 standard deviation could become negative. I suggest to cancel this section or to better argument its goal and procedure.

Section 4.2. In many cases, there are footprints where the raingauges do not measure rain, while gridded value is above zero. In some cases (e.g. footprints 2, 5, 8, 11 and 13 of event 9) eight or nine gauges measure zero, but interpolation fills the footprints with not negligible amount of rain. The Authors should justify the use of the interpolated data.

Page 16, Line 5. The convective/stratiform discrimination can be done in several ways (see Bringi, et al., 2003, J. Atmos. Sci. 60, 354–365 for an example). How it is done here? It is quite strange that light precipitation belongs to convective events. . .

Page 16, Line 10. Are the DPR footprints in chronological order? In terms of milliseconds?

Page 19, line 5 to the end. I understand the point (see my previous comment), but if the lag is supposed to be due to the time needed to raindrops to reach the ground, it is largely overestimated here. A raindrop of 2 mm of diameter has a terminal velocity around 6 m/s, that means that in 5 minute it covers around 1800 m, and I guess the DPR near surface bin is closer to the ground. Thus to search a time lag larger than 5-10 minutes is not justified. The relatively higher correlation found at lag of 10-20 minutes are very likely due to the patchy rain pattern and to the very small rainrates.

Conclusions.

Page 20, Line 13. Probably gridded data add information, but there is no guarantee that the information is correct.

---

## Referee Comment (RC2) · Anonymous Referee #2 · 20 Dec 2018

**General comments**

The paper presents an evaluation of precipitation estimates by the DPR onboard the GPM Core Satellite by using a very dense gauge network located in Austria. The study can be of interest for the GPM DPR user community, because it aims at providing an independent validation of DPR precipitation estimates. Anyway many issues are present in the manuscript, that are worthwhile of further investigation. First of all the 22 selected case studies include many very light precipitation events, or no precipitation at all. The high number of correct negatives has a strong impact on the statistics and should be not included in the analysis. Moreover I do not see an evaluation of the rainfall variability in the DPR footprint, thanks to the very dense gauge network. Thus the usefulness of such a density seems to be not fully exploited. English can be

improved. Many typos are present and therminology used is often quite approximative and needs to be checked. Moreover the statistical analysis is not well designed in my opinion. At the end my feeling is that the paper does not add enough relevant and new knowledge on the topic stated in the title. Thus I think the paper could be considered for publication in AMT, provided that a careful effort of major revision is undertaken.

**Specific comments**

Page 2 Line 18: If by "The microwave imagers" you mean the GMI, please reword "The microwave imagers augment the core satellite and enable a high temporal resolution for global precipitation maps." to "The GMI completes the core satellite, enables a high temporal resolution for global precipitation maps and is used as a calibrator for the other radiometers in the constellation." If you are referring to the other radiometers of the GPM constellation, please use the words "microwave radiometers", not " microwave imagers", because not all of the constellation instruments are imagers (e.g. MHS is a sounder).

Page 3 line 2: "only measurements at points in time" means "instantaneous precipitation rate"

Page 3 lines 4-5: more and better with respect to?

Page 4 line 12: what do you mean by "up to 0.1 mm"?

Page 4 lines 19-28: I do not agree with the authors about this paragraph. Why do they say that the WegenerNet is twice as good as the DPR? The availability of about 1 gauge per 2 km2, while the DPR resolution is roughly 5x5 km2, makes the difference in resolution much higher. Moreover they say "there is no other precision (quality) information for the GPM-DPR estimates": what do they exactly mean?

The sentence "the WegenerNet provides accurate information, not only whether precipitation occurred but also on the amount of rain." seems to say: the DPR only provides where precipitation occurred. Please clarify better. AMTD
Finally: I do not think that the gridded gauge data are actually characterized by a higher resolution, with respect to the the station data. I think that the inverse distance method used to regrid data cannot really "increase" the resolution from 2 km2 to 200m x 200 m. Please explain better this issue.

Page 5 lines 5-10. The concept is often right, but this paragraph should be written in a more precise way. The DPR swath includes 49 beams (or rays), not bins. The KaPR includes 49 beams in total, but only 25 are overlapped to the KuPR ones, the other 24 are interlaced. The sentence "The KaPR on the other hand, has half of the swath size of KuPR with 120 km and 49 bins" seems to mean that KaPR has roughly twice the KuPR resolution. The sentence "KaPR shall provide better information on light rainfall and snow." is not completely correct. This was the aim, in some sense, of the availability of Ka-band in the DPR, with respect to TRMM PR I mean. Anyway there is a relevant bibliography dealing with the problem of detecting light rainfall and especially snow by means of KaPR, because of its low sensitivity (e.g. Casella et al, that is the list of references, but is not cited in the manuscript).

Pag 5 line 14-15: DPR does not measure cumulated rainfall, but instantaneous one. So here you should write 0.2 mm/h and 0.5 mm/h.

Page 9 lines 23-24: because the authors are aware that correct negatives take an effect on the results, why do they include them in the statistical analysis. I should suggest to avoid them and to use other scores (e.g. ETS, HSS) to evaluate the performances. In which way you carefully choose the events?

Page 11 lines 6-12: This paragraph is not clear. I do not understand how you round to zero. The analysis of the subpixel-scale variability is cited, but non investigated in details throughout the manuscript.

Pages 12-14: all this part is not clear and in general shoul be rethought. Moreover the last analysis, with the constraint on GPM-DPR estimates and standard deviation, could be completely avoided.

AMTD
Section 4.2 The analysis of case studies is misleading. How can it happen that for some footprint stations do not measure precipitation and gridded data show a (relevant) amount of rainfall? It often happens and the authors shoul explain if the inverse distance used to regrid data is responsible for this strange behaviour. If so, I do not think that the gridded data have to be used, also because of the artificial higher resolution.

Page 16 lines 1-6. Light rain are of convective nature? I cannot understand how you discriminate between convective and startiform events.

Page 19 Lines 6-8: the analysis of the lag effect is correct in principle, but I do not understand the use of such a large time interval (+ 30 minutes).

Page 20 Line 6: what does "although the grid changed obviously" mean?

Page 21 Lines 12-14: I do not understand what are the authors' plans as far as HS scan is concerned.

**References**

Many items present in the list of references (e.g. Casella et al., Petracca et al., Seto et al., Speirs et al., Szeberényi et al.) are actually not cited in the manuscript. Please check carefully.

In the list of references please write the author names in the right way and put them in the right alphabetical order (e.g. Jackson is actually Skofronick-Jackson)

**Technical corrections**

There are many typost hrought the manuscript. Just some of them are listed below. The authors have to check very carefully all of them.

Page 2 Line 8: is the reference to JAXA in the correct AMT style?

Page 3 line 8: fly-overs are generally called overpasses.
Page 4 line 19: "each footprint"  $\rightarrow$  "each DPR footprint"

Page 5 line 1: "frequency" here is really misleading. Maybe you mean "number, concentration"?

Page 6 line 5: the most rainiest  $\rightarrow$  the rainiest.

Page 10 line 1: occurs→occur

Page 10 line 25: any $\rightarrow$ no?

Page 13 line 16: "delivers"  $\rightarrow$  "deliver"

Page 13 line 17: "low"  $\rightarrow$  "light"

Page 16 line 3: phenomena is plural

Page 16 line 1: "where"  $\rightarrow$  "were"

Page 17 line 1: Fig A6 is actually Fig A5.

Page 18 line 3: Fig A7 is actually Fig A6.

Page 19 line 17: Fig A5is actually Fig A4.

Figures

Figure 1: Place the label somewhere else.

Figure 3: Does min resolution Ka (that is 0.2 mm/h non t mm) need to be shown?

Figure 4 caption: what is resp.? "The diagonal denotes the line where the satellite measures the same as the terrestrial network." is obvious.

Figure 5 top panel is the same as Figure 3. Please avoid to show the same data twice.

Figure 6-9. The DPR data superimposed to the Wegener gridded ones in the bottom panel should be removed, because already shown in the top panel.

AMTD

---

## Referee Comment (RC3) · Anonymous Referee #3 · 3 Jan 2019

Review of AMT-2018-395 By Martin Lasser, Sungmin O and Ulrich Foelsche. Manuscript title: Evaluation of GPM-DPR precipitation estimates with WegenerNet gauge data.

The manuscript presents a validation of the DPR rain rate products by using as a reference the rain gauge data collected by the WegenerNet network in the South-East Austria. Four years of data have been considered, but due to the small spatial extension of the network only 22 case studies have been selected. Both dichotomous (i.e. POD, FAR) and continuous (i.e. Bias, Correlation coefficient) scores have been calculate in order to quantify the agreement between satellite and ground-based data. The GPM community encourages validation studies and in this sense the paper is of absolute interest. However, in the current shape it presents some confusing parts and,

even if it is easy to read and to be followed, the English can be improved. As a general comment, the paper presents a lack of the GPM validation works which have already been carried out (i.e. Watters et al., 2018, D'Adderio et al., 2018., Biswas and Chandrasekar, 2018 among the other). Some of them are reported in the references list but are never cited within the text (Petracca et al., Speirs et al. – also other papers are not cited within the text). The review of the works on the same topic is useful to understand the novelty of the present work. In this regard, in my opinion, the authors do not take advantage of a such dense gauges network. There is not any analysis relating the agreement between DPR products and the rain gauges measurements to the intra-footprint variability. This would considerably increase the quality of the paper. Following these general comments and the more specific ones reported below, the manuscript has to be deeply revised before to be considered suitable for publication on the Atmospheric Measurement Technique journal.

- Page 2, lines 13-15 and lines 22-23: these information should be moved in the Section 2.

- Page 4, line 30: "The DPR radar instrument..." should be "The DPR instrument..."

- Page 4 ,lines 30-32: the heavy and light precipitation can occur also at mid- and tropical latitudes.

- Page 5, lines 7-10: I see what the authors mean saying that the KaPR has 49 bins (25 of MS + 24 of HS), but I think this is misleading for a user which does not have a deep knowledge of DPR data structure. I suggest to rewrite the period also considering that the Ka HS data are not used in this study.

- Page 7, lines 1-3: I do not agree with the choice of the authors to include the three events where the DPR products did not reported rain. This is in contrast with what they state at lines 25-27 pf Page 5.

- Page 7, lines 3-6: what is the reason to report an event in Table 1 and then state that

the same event had to be omitted?

- Page 7, lines 13-14: not all the plots of Figure 2 report the WegenerNet rain rate map, so you should specify which panels you refer. In general, it would be very useful to label each panel with a), b), c) and so on. Furthermore, the bottom panels of Figure 2 are useless, the panel titles "WegenerNet grid" is enough.

- Page 9, line 2: has the event N. 21 of Table 1 been omitted or not?

- Page 9, lines 12-17: all the data available are included in the calculation of both correlation coefficient and bias. This means to include also the DPR-gauge couple both reporting no rain. Consequently, this has an effect (as also the authors state) on the scores. You should to consider only the data where the truth (gauges) reports rain rate above the minimum detectable threshold (1 mmh-1) for the calculation of correlation coefficient and bias.

- Page 10, line 25 to Page 11, line 2 and Figure 3 and Figure A3: this is totally confusing to me. What is the difference between the blue line of Figure 3 and Figure A3? If the Figure A3 reports the range it should have a minimum and a maximum value for each footprint, but I see only one value. Furthermore, the standard deviation reported as it is in Figure A3 is completely useless. You should plot the average value plus/minus one standard deviation represented by a shadowed area.

- Table 3 should be placed before than Table 4.

- Discussion about contingency table (Table 3) and derived statistics (Table 4): in my opinion, it is hard to say that KaPR provides the best results since it reports a lot of misses (the double of KuPR and DPR) which are not take into account considering only POD, FAR and POFD. Consequently, I suggest to include additional statistical scores (as for example, the Critical Success Index – CSI) to evaluate the impact of the misses.

- Figure 5: the top panel is repeating the information given in Figure 3. It could substitute the Figure 3 or it should be deleted. Where is the CC for the event N. 8, 10 and

СЗ

13? Furthermore, some events do not show the CC for all three DPR products. Can you explain the reason?

- Page 15, lines 9-10: instead of the mean of CC, you could consider to calculate the CC by combining all the data available.

- Page 15, line 11: it does not make any sense to calculate the bias for the events without precipitation.

- Page 16, line 2: substitute "where" with "were".

- Page 16, lines 18-20: which constraint?

- Within the text, the values of statistical scores are reported sometimes as absolute values and sometimes as percent values, while they are reported in absolute values in the corresponding tables. Please, uniform the text reporting either absolute or percent values.

- Page 18, line 8: I do think that rain rate lower than 7 mmh-1 can be considered heavy precipitation.

- Page 19, line 1: substitute "events" with "event".

- Page 19, line 6-8: a time lag of  $\pm$ 30 minutes is too large. The DPR precipitation estimation at surface that you used is an extrapolation of the precipitation rate estimated at Near Surface level, which is the level closest to the surface where the data are available. Considering the distance between this level and the surface you can estimate the time needed to the precipitation to reach the ground and consider a reasonable time lag.

Please also note the supplement to this comment: https://www.atmos-meas-tech-discuss.net/amt-2018-395/amt-2018-395-RC3supplement.pdf Interactive comment on Atmos. Meas. Tech. Discuss., doi:10.5194/amt-2018-395, 2018.

---

## Author Comment (AC1) · 12 Mar 2019

**Reply to the Comments from Referee #1 for AMT-2018-395**

*We would like to thank the referee for the review of our paper and the constructive comments. In the following, we have provided an item-by-item reply to the comments.*

**Major Comments:**

1) The paper is fairly well written, and the topic falls within the scope of the journal. I feel, however, that experiment is not well designed for two main reasons. First, the accumulation time selected for the raingauges is too short. If I well understood, the sensitivity of the TB is 0.1 mm, so that any measure X has to be intended as XïC´s0.1 mm. By considering such short time, most of the time (given the low rainrates measured in the area) the data should be (looking at figures 6 to 9) for a single gauge 0.1 mm, resulting in a value of 1.2ïC´s1.2mm/h, that means a relative error of 100% for the most frequent value. Tokay et al., (2003, J. Atmos. Oceanic Technol., 20, 1460-1477) shows that for very light rain amount, the correlation between co-located tipping bucket raingauges can be very low. I suggest to use longer raingauge cumulation intervals (see also Porcù et al., 2014, Atmospheric Research) and to discuss the error associated to the ground measure.

> *Considering the accumulation time, which easily introduces errors to the evaluation, we refer to Tan et al. (2018) (https://journals.ametsoc.org/doi/full/10.1175/JHM-D-17-0174.1): "To extract the precipitation rate from the gauges, we will need to select an accumulation time. On one hand, since the satellite retrievals are considered instantaneous, the gauge accumulation time should be as short as possible. On the other hand, gauges have measurement uncertainty, so the gauge accumulation time should be sufficiently long to ensure a "stable" gauge measurement. After some trial and error, we determine 5 min to be a reasonable balance between these two factors."*

> *Furthermore, we show the DPR performance with lagged (+- 30 min) gauge data (Fig. 10 – which will be updated to +- 15 min) based on the assumption that Level 2 DPR is supposed to provide only a snapshot of rainfall data. Thus, we selected the shortest accumulation data.*
> *As a further example Amitai at al. (2012) (https://journals.ametsoc.org/doi/full/10.1175/JHM-D-12-016.1) also follow the approach of using 1 minute gauge data to compare with satellite radar data including a time lag instead of using accumulated gauge data.*

2) The second issue is on the use of binary indicators to assess the quality of the DPR estimates. Two indicators (PC and POFD) includes the number of correct negatives as input. This number should not be considered in the evaluation, since can be arbitrarily larger or smaller by changing the selection of cases, and thus the results are not general. Moreover, I suggest to use other indicators (ETS, HSS, HK) to synthetize POD and FAR information

*We agree, the PC and POFD cannot be generalised since they are heavily dependent on the sample selection. We had chosen to incorporate them because the number of footprints showing precipitation almost equals the number without precipitation. However, as you pointed out the dependency on the correct negatives leads to misinterpretations. We will therefore cancel the PC and the POFD and move to ETS and CSI as well as HSS. We started the computations and will include them in the revised manuscript.*

**Specific Comments:**

1) Page 2, Lines 18-19. The imager is only one (GPM Microwave Imager) and it is designed to provide a radiometric standard for the other radiometers of the GPM constellation.

   *Indeed, this sentence referred erroneously to the microwave instruments (microwave-radiometer, -imager, -sounder) of the constellation satellites. We change it to "The GMI completes the core satellite, enables a high temporal resolution for global precipitation maps and is used as a calibrator for the other radiometers in the constellation.". Sorry for the confusion.*

2) Page 3, Line 1. There are two recent papers (Speirs et al, 2017; Petracca et al., 2018) performing a similar analysis in the same region (Switzerland and Italy): they are reported in the reference list but not mentioned in the paper. Also the important paper Seto and Iguchi (2014) is in the reference but not mentioned in the text. The Authors should carefully read these papers and discuss their results.

   *Thank you for those remarks, we will incorporate them correctly.*

3) Page 4, Lines 12-15. No solid precipitation in summer (i.e. hail)?

   *You are right; the formulation was not precise enough. Fortunately, the events in this study do not incorporate a hail event. In general, hail events affect only very few stations, which are then treated by the quality control of the WegenerNet like those that are clogged (e.g. by leaves). Their data are not taken into account in the calculation of the gridded product. In the entire WegenerNet time series there is just one event, where three stations have been simultaneously affected by hail.*

4) Page 4. Line 27. Inverse distance interpolation does not add information to the gridded data, since the only information is in the raingauge measurements. The increased resolution can be good or bad, depending on how the precipitation pattern agrees with the inverse distance model.

   *This is true, but the interpolated data can include rainfall information from gauges that are located outside of DPR footprints but still within a radius of influence. Additionally, in most cases there is very little difference between gauge and gridded WegenerNet data (see Figure A1/A2). We think that the better coverage of the gridded data delivers more reliable results (see event no. 9).*

5) Page 5, Line 1. Frozen hydrometeors attenuate radiation as well.

*Thank you for careful reading, we will add that.*

6) Page 5, Lines 19-20. It has to be considered that the radar measures precipitation in a volume at a given altitude (near surface bin, I guess), and it takes time to the raindrops to reach the ground. In case the Authors claim for a very precise temporal matching, this issue should be discussed. It should also be mentioned if the "near surface bin" value is used to evaluate the vertical distance between the DPR estimate and the ground.

*We wanted to point out the challenge here. However, we used the precipitation rate for the estimated surface, which definitely needs to be mentioned. This will be done in section 2.3 "Selected data". The effect of potential time lags is addressed in Fig. 11 (within +- 30 min [Note that this figure will be updated to show just +- 15 min]).*

7) Page 7, Line 12.It is "8 to 10" or 8-12 (Page 6, Line 1) stations for footprint?

*It is "8-12 stations". Thank you for the hint.*

8) Page 9, Lines 15-16. Bias and Normalized Bias give different information and should be computed both. The fact that the NB gives huge values simply tells that the error is much larger than the measurement, this is often the case when a too short cumulation time is used (5 min).

*Indeed, in case of a very small mean, the normalised bias gives information that it is hardly possible to detect anything in this event. We follow your suggestion and will show both the bias and the normalised bias in the revised manuscript.*

9) Page 9, Lines 19-22. To build any contingency table the threshold to define rain and no-rain sample has to be carefully defined and reported here.

*The threshold was set to 0 mm/5min. We will report that. A change of the threshold to 0.1 mm/5min has little impact. The contingency table for KuPR is then:*
*92 hits - 47 false al. instead of 95 hits - 44 false al.*
*28 misses - 86 corr. neg.     31 misses - 83 corr. neg.*

10) Page 9, Lines 23-24. As mentioned, the direct use of correct negatives should be avoided in any validation study (see Nurmi, 2003, Recommendations on the verification of local weather forecasts. ECMWF Technical Memoranda. Technical Memorandum No. 430 for reference). What is the precise meaning of the "careful choice" of the events? For this reason the indicator PC and POFD should be removed by the analysis.

*Starting from the end of your comment: The "careful choice" means that we found a balanced environment of rain/no-rain events. However, we will remove PC and POFD, to avoid the usage of correct negatives. Consequently, the "careful" choice can and will be omitted as well.*

11) Page 10, Line 21. A key indicator is the Equitable Threat Score (see Nurmi, 2003) that summarizes both POD and FAR, and gives the skill with respect to the random assignment of rainy footprint.

   *We will include the ETS and the Critical Skill Score (CSI) to score the rain events.*

12) Page 10, Line 25-26. This sentence is useless and should be cancelled.

   *Cancelled.*

13) Page 10, Line 27. What is the "general structure"?

   *We intended to state a broad agreement between the DPR and the WegenerNet.*

14) Page 10, Lines 28. What does it mean "70% of the GPM-DPR precipitation rates are within the range of the respective WegenerNet gauges"? What is the "range of WegenerNet gauges"?

   *I. e. the interval [min_value, max_value] of the WegenerNet within the respective footprint. For 70% of the DPR rain rates ($r_{DPR}$) $r_{DPR} \in$ [min_value, max_value] holds.*

15) Page 11, Line 7. What does it mean "close to zero"? How are the numbers rounded? How many significant digits are considered?

   *One digit after the decimal point is considered, which means zero = 0.0 mm/h. The "rounded to zero" will be omitted, that was for the visual interpretation.*

16) Page 12, Lines 8-10. Please avoid misleading numbers and cancel PC and POFD from the analysis.

   *We will skip them as addressed above.*

17) Page 13, Lines 1-4. FBI gives a measure of the under-/overestimation of the wet area, not of precipitation.

   *Thank you, this is a neat explanation, we will include it.*

18) Page 14, Lines 1-end. I'm not sure this analysis is well designed. If I understood well, hit is when the DPR products falls within +- 1 standard deviation of the corresponding gauges value, misses is if the DPR is lower than the gauges – 1 standard deviation, and false alarm if the DPR is higher than the gauges+1standard deviation value. Who are the correct negatives? How large is the standard deviation (roughly)? The main problem I see is that the distribution of rainrates is strongly asymmetric (power law) so that the standard deviation is asymmetric with respect to the mean value. In case of very light rainrate the value gauge-1 standard deviation could become negative. I suggest to cancel this section or to better argument its goal and procedure.

*As this analysis is confusing to everyone, we cancel it from the manuscript.*

19) Section 4.2. In many cases, there are footprints where the raingauges do not measure rain, while gridded value is above zero. In some cases (e.g. footprints 2, 5, 8, 11 and 13 of event 9) eight or nine gauges measure zero, but interpolation fills the footprints with not negligible amount of rain. The Authors should justify the use of the interpolated data.

*We use the gridded data because the DPR represents areal rainfall within its footprint while station data are only point-data. Thus, the interpolation increases the spatial resolution and can include rainfall information from gauges that are located outside of DPR footprints but still within a radius of influence. This is one of the advantages of a dense network.*

20) Page 16, Line 5. The convective/stratiform discrimination can be done in several way (see Bringi, et al., 2003, J. Atmos. Sci. 60, 354–365 for an example). How it is done here? It is quite strange that light precipitation belongs to convective events.

*This is an error in the manuscript, it is exactly vice versa than written. Sorry for the confusion.*

21) Page 16, Line 10. Are the DPR footprints in chronological order? In terms of milliseconds?

*Yes, they are. Ordered as given in the data. However, in order not to confuse any reader we delete the reference to the chronological order, since it is not important for our analyses. "The precipitation comparison at each footprint is given in Fig. 6"*

22) Page 19, line 5 to the end. I understand the point (see my previous comment), but if the lag is supposed to be due to the time needed to raindrops to reach the ground, it is largely overestimated here. A raindrop of 2 mm of diameter has a terminal velocity around 6 m/s, that means that in 5 minute it covers around 1800 m, and I guess the DPR near surface bin is closer to the ground. Thus to search a time lag larger than 5-10 minutes is not justified. The relatively higher correlation found at lag of 10-20 minutes are very likely due to the patchy rain pattern and to the very small rainrates.

*This describes in short what we found as well. The large lag time was not only to show a potential time lag caused by the fall velocity of a raindrop but also to have a look at the patterns that might arise (especially of interest in case of light rain). However, since this obviously causes confusion, we decided to restrict the lag-window to 15 minutes.*

23) Conclusions. Page 20, Line 13. Probably gridded data add information, but there is no guarantee that the information is correct."

*Please refer to the answer in specific comment (19).*

---

## Author Comment (AC2) · 12 Mar 2019

**Reply to the Comments from Referee #2 for AMT-2018-395**

*We would like to thank the referee for the review of our paper and the constructive comments provided. In the following, we have provided an item-by-item reply to the comments.*

**Major Comments:**

1) The paper presents an evaluation of precipitation estimates by the DPR onboard the GPM Core Satellite by using a very dense gauge network located in Austria. The study can be of interest for the GPM DPR user community, because it aims at providing an independent validation of DPR precipitation estimates. Anyway many issues are present in the manuscript, that are worthwhile of further investigation. First of all the 22 selected case studies include many very light precipitation events, or no precipitation at all. The high number of correct negatives has a strong impact on the statistics and should be not included in the analysis. Moreover I do not see an evaluation of the rainfall variability in the DPR footprint, thanks to the very dense gauge network. Thus the usefulness of such a density seems to be not fully exploited. English can be improved. Many typos are present and therminology used is often quite approximative and needs to be checked. Moreover the statistical analysis is not well designed in my opinion. At the end my feeling is that the paper does not add enough relevant and new knowledge on the topic stated in the title. Thus I think the paper could be considered for publication in AMT, provided that a careful effort of major revision is undertaken.

> *You are right, the number of correct negatives influences the statistics, even though the amount of rain/no-rain events is balanced. In order not to provide misleading numbers, we will therefore exclude these biased indicators (PC and POFD). Insetad, we will include the ETS, CSI and HSS to provide reasonable statistics. As the rainfall variability is not sufficiently addressed we will discuss the intra-footprint variability based on a scatter plot of |WegenerNet – DPR| and std(WegenerNet). The inter-footprint variability will be approached by avg(|WegenerNet – DPR|) in one event and std(WegenerNet) in one event. Terminology and English will be checked and improved.*

**Specific Comments:**

1) Page 2 Line 18: If by "The microwave imagers" you mean the GMI, please reword "The microwave imagers augment the core satellite and enable a high temporal resolution for global precipitation maps." to "The GMI completes the core satellite, enables a high temporal resolution for global precipitation maps and is used as a calibrator for the other radiometers in the constellation." If you are referring to the other radiometers of the GPM constellation, please use the words "microwave radiometers", not " microwave imagers", because not all of the constellation instruments are imagers (e.g. MHS is a sounder).

> *Thank you for the correction and phrasing. We are happy to use your nice wording, that is exactly what we wanted to say.*

2) Page 3 line 2: "only measurements at points in time" means "instantaneous precipitation rate"

*Thank you for the clarification, which will be incorporated.*

3) Page 3 lines 4-5: more and better with respect to?

   *It was meant to be with respect to the GPM-DPR measurements.*

4) Page 4 line 12: what do you mean by "up to 0.1 mm"?

   *The bucket has a volume of 0.1 mm equivalent.*

5) Page 4 lines 19-28: I do not agree with the authors about this paragraph. Why do they say that the WegenerNet is twice as good as the DPR? The availability of about 1 gauge per 2 km2, while the DPR resolution is roughly 5x5 km2, makes the difference in resolution much higher. Moreover they say "there is no other precision (quality) information for the GPM-DPR estimates": what do they exactly mean? Finally: I do not think that the gridded gauge data are actually characterized by a higher resolution, with respect to the the station data. I think that the inverse distance method used to regrid data cannot really "increase" the resolution from 2 km2 to 200m x 200 m. Please explain better this issue.

   *Twice as good is referring to the minimum resolution in the amount of rain, not to the areal resolution, but you are right, this is a "non-scientific" phrasing. The sentence "there is no other precision (quality) information for the GPM-DPR estimates", had the intention to state that the DPR products do not deliver a value to assess the quality of each estimate (e. g. the standard deviation). The interpolation does indeed not add new information, but allows to include rainfall information from gauges that are located outside of DPR footprints - but still within a radius of influence. This is one of the advantages of a dense network. We rephrase that paragraph for more clarity.*

6) Page 5 lines 5-10. The concept is often right, but this paragraph should be written in a more precise way. The DPR swath includes 49 beams (or rays), not bins. TheKaPR includes 49 beams in total, but only 25 are overlapped to the KuPR ones, the other 24 are interlaced. The sentence "The KaPR on the other hand, has half of the swath size of KuPR with 120 km and 49 bins" seems to mean that KaPR has roughly twice the KuPR resolution. The sentence "KaPR shall provide better information on light rainfall and snow." is not completely correct. This was the aim, in some sense, of the availability of Ka-band in the DPR, with respect to TRMM PR I mean. Anyway there is a relevant bibliography dealing with the problem of detecting light rainfall and especially snow by means of KaPR, because of its low sensitivity (e.g. Casella et al, that is the list of references, but is not cited in the manuscript).

   *Thank you very much for this comment. We will rewrite that for more clarity. The use of "bins" stems from the GPM Data Utilization Handbook (third edition), where the beams are referred as "angle bins". It was not meant to be mixed up with the range bins within one angle bin.*

7) Page 5 line 14-15: DPR does not measure cumulated rainfall, but instantaneous one. So here you should write 0.2 mm/h and 0.5 mm/h.

*Thank you for the remark, we will change it accordingly.*

8) Page 9 lines 23-24: because the authors are aware that correct negatives take an effect on the results, why do they include them in the statistical analysis. I should suggest to avoid them and to use other scores (e.g. ETS, HSS) to evaluate the performances. In which way you carefully choose the events?

*The "careful choice" means that we found a balanced environment of rain/no-rain events. Following your suggestion, we will cancel the PC and the POFD to avoid the usage of correct negatives and move to ETS and CSI as well as HSS. Consequently, the "careful" choice can be omitted as well.*

9) Page 11 lines 6-12: This paragraph is not clear. I do not understand how you round to zero. The analysis of the subpixel-scale variability is cited, but non investigated in details throughout the manuscript.

*One digit after the decimal point is considered, which means zero = 0.0 mm/h. The "rounded to zero" will be omitted, that was for the visual interpretation. As the rainfall variability is not sufficiently addressed we will discuss the intra-footprint variability based on a scatter plot of |WegenerNet-DPR| and std(WegenerNet).*

10) Pages 12-14: all this part is not clear and in general should be rethought. Moreover the last analysis, with the constraint on GPM-DPR estimates and standard deviation, could be completely avoided. Section 4.2 The analysis of case studies is misleading. How can it happen that for some footprint stations do not measure precipitation and gridded data show a (relevant) amount of rainfall? It often happens and the authors shoul explain if the inverse distance used to regrid data is responsible for this strange behaviour. If so, I do not think that the gridded data have to be used, also because of the artificial higher resolution.

*We delete the analysis with the constraint of the standard deviation. The interpolated gridded data can include rainfall information from gauges that are located outside of DPR footprints but still within a radius of influence.*

11) Page 16 lines 1-6. Light rain are of convective nature? I cannot understand how you discriminate between convective and startiform events.

*We are sorry for the confusion. In fact it is vice versa than written in the paper.*

12) Page 19 Lines 6-8: the analysis of the lag effect is correct in principle, but I do not understand the use of such a large time interval (+ 30 minutes).

*The large lag time was not only to show a potential time lag referring to an error in the estimated surface bin but also to have a look at the patterns that might arise*

*(especially of interest in case of light rain). However, since this obviously causes confusion, we decided to restrict the lag-window to 15 minutes.*

13) Page 20 Line 6: what does "although the grid changed obviously" mean?

*We wanted to draw the attention to the fast moving core of the rainfall event, which is reflected in the correlation. The explanation for the correlation peak by the time of the measurement is the only valid one.*

14) Page 21 Lines 12-14: I do not understand what are the authors' plans as far as HS scan is concerned.

*We will rephrase the last paragraph and cancel the HS-plans as it is off the track. We are sorry for the confusion.*

**References:**

Many items present in the list of references (e.g. Casella et al., Petracca et al., Seto et al., Speirs et al., Szeberényi et al.) are actually not cited in the manuscript. Please check carefully. In the list of references please write the author names in the right way and put them in the right alphabetical order (e.g. Jackson is actually Skofronick-Jackson)

*Thank you for the remarks. There was a misunderstanding concerning correct citations. We revise the manuscript for this deficiency.*

**Technical corrections:**

There are many typos throught the manuscript. Just some of them are listed below. The authors have to check very carefully all of them.

*We checked the manuscript carefully and found quite some typos. Thank you for the kind remarks, we cleaned up the mess, deleted duplicates and performed the changes as suggested.*

Figures
    Figure 1: Place the label somewhere else.
    Figure 3: Does min resolution Ka (that is 0.2 mm/h non t mm) need to be shown?
    Figure 4 caption: what is resp.? "The diagonal denotes the line where the satellite measures
          the same as the terrestrial network." is obvious.
    Figure 5 top panel is the same as Figure 3. Please avoid to show the same data twice.
    Figure 6-9. The DPR data superimposed to the Wegener gridded ones in the bottom panel
          should be removed, because already shown in the top panel.

*Thank you for all those remarks. The information that is shown twice was just for easy reading. If better, we remove it. The "resp" in the caption of Fig. 4 was meant to be*

*respective, which indeed needs to be written in full. We adapt the figures according to the remarks.*

---

## Author Comment (AC3) · 12 Mar 2019

Dear Reviewer, we thank you very much for your careful review and the helpful comments. Please find our item-by-item reply in the supplement section.

Please also note the supplement to this comment:
https://www.atmos-meas-tech-discuss.net/amt-2018-395/amt-2018-395-AC3-supplement.pdf

---

## Referee Report (RR1)

Review of AMT-2018-395 – Revised Version 1

By Martin Lasser, Sungmin O and Ulrich Foelsche.

Manuscript title: Evaluation of GPM-DPR precipitation estimates with WegenerNet gauge data.

The revised version of the manuscript presents significant changes with respect to the original version. The authors mainly addressed the reviewers' comments, even if I am not totally convinced about the significance of new analyses. I particularly appreciate the analysis on the relationship between the DPR-Wegener error and the intra- and inter-footprint variability. At the same time, I retain that using the Wegener grid standard deviation to characterize the variability is not the better choice. Figure 4 shows standard deviation values generally higher than the mean rainfall rate measured by the rain gauges (Table 1) and comparable to the mean rainfall rate of most of the footprints (Figures 6-9). In their discussion, the authors state that there is no correlation between the intra-footprint variability and the error between DPR and Wegener. In my opinion, they should repeat the analysis with the coefficient of variation instead of the standard deviation to have a confirmation of what they assert. Furthermore, Figure 4 shows some points on the y-axis, that is standard deviation equal to zero. Which situation do they describe? Does the precipitation is uniform and higher than zero in each grid-point or is uniform but equal to zero in each grid-point?

Another focal point, is the calculation of binary statistical scores (i.e. POD, FAR, HSS, etc.) and the choice to include any positive rainfall rate. This could be acceptable for the rain gauges because their measure the precipitation, but it cannot be accepted for the DPR because it estimates the rainfall rate. The authors report in the manuscript the minimum resolution of both DF and Ka-/Ku-only algorithms (according to official GPM documents), but they not consider this condition in their analyses. They could even double check with the reflectivity measured by DPR that should be above the minimum detectable signal. This could further change the results shown in Tables 3 and 4.

These are the two main points that, together with minor points listed below, have to be correctly addressed before to retain the manuscript printable on Atmospheric Measurement Technique journal.

- Page 5, lines 8-9: the KaPR has 25 beams as the Matched Scan (MS).

- Page 19, line 6: I would say "…featuring light to moderate precipitation, up to almost 6 mmh$^{-1}$." From Figure 9, I do not see any points with rainfall rate higher than 6 mmh$^{-1}$.

- Figures 6-9: sometimes there is not spatial match between the Wegener stations and Wegener grid. For example, the sixth box of Figure 7 shows that the grid points closer to the stations with rainfall rate around 1.5 mmh$^{-1}$ report almost zero rainfall rate , while the grid point farer from the stations report rainfall rate higher than 1 mmh$^{-1}$. Can you explain why?